# Portable Reward Tuning:
# Towards Reusable Fine-Tuning across Different Pretrained Models

**Daiki Chijiwa** [* 1]  **Taku Hasegawa** [* 2]  **Kyosuke Nishida** [2]  **Kuniko Saito** [2]  **Susumu Takeuchi** [1]

## Abstract

While foundation models have been exploited for various expert tasks through fine-tuning, any foundation model will become outdated due to its old knowledge or limited capability. Thus the underlying foundation model should be eventually replaced by new ones, which leads to repeated cost of fine-tuning these new models. Existing work addresses this problem by inference-time tuning, i.e., modifying the output probabilities from the new foundation model with the outputs from the old foundation model and its fine-tuned model, which involves an additional overhead in inference by the latter two models. In this paper, we propose a new fine-tuning principle, Portable Reward Tuning (PRT), that reduces the inference overhead by its nature, based on the reformulation of fine-tuning as the reward maximization. Specifically, instead of fine-tuning parameters of the foundation models, PRT trains the reward model explicitly through the same loss function as in fine-tuning. During inference, the reward model can be used with any foundation model (with the same set of vocabularies or labels) through the formulation of reward maximization. Experimental results, covering both vision and language models, demonstrate that the PRT-trained model can achieve comparable accuracy to the existing work of inference-time tuning, with less inference cost.

## 1. Introduction

Foundation models, or simply pretrained models, play a central role in the recent development of artificial intelligence (Bommasani et al., 2021). They are typically large-scale neural networks pretrained on massive amounts of data from the Internet, which makes them generalizable to various downstream tasks, such as CLIP (Radford et al., 2021) for visual recognition and LLAMA-series (Touvron et al., 2023a;b) for language generation. Even though foundation models are somewhat already capable of handling downstream tasks, we can further bring out their potential ability by *fine-tuning* for each task, i.e., additional optimization of the foundation models on supervised data. Furthermore, fine-tuning can also be used for aligning the behavior of foundation models to follow human instructions or preferences, by instruction tuning or reinforcement learning from human feedback (RLHF; Christiano et al. (2017)).

Although fine-tuning has been the standard principle for tuning foundation models, there still remains an overlooked problem in the long term: the underlying foundation models should be eventually replaced by other (often newer) foundation models for various reasons, such as their outdated knowledge or limited capability, which requires the new foundation models to be fine-tuned on every task again. For example, (i) in the case of open-source CLIP models (Schuhmann et al., 2021; 2022), an early model was initially trained on a dataset with 400 million image-text pairs, but later it was replaced by new versions trained on 2 billion and 5 billion pairs; (ii) in the case of the LLAMA series, both major updates (e.g., LLAMA-2 to LLAMA-3) and minor updates (e.g., LLAMA-3.1 to LLAMA-3.2) have occurred. Every time such an update occurs, one may have to fine-tune the new foundation model again, which poses an important issue of the repeated training cost.

One promising approach for this challenge is *inference-time tuning* (Mitchell et al., 2024; Liu et al., 2024a), which emulates fine-tuning of the new foundation model even with a possibly different architecture, by mixing three output probabilities from the old foundation model, its fine-tuned model, and the new foundation model. More specifically, the approach first defines an *implicit reward* for each task as the logarithmic density ratio of the output probabilities from the (old) foundation and its fine-tuned models. Then, it reinterprets fine-tuning as maximization of the implicit reward with penalizing the deviation from the foundation

---

[*]Equal contribution  [1]NTT Computer and Data Science Laboratories, NTT Corporation  [2]NTT Human Informatics Laboratories, NTT Corporation. Correspondence to: Daiki Chijiwa <daiki.chijiwa@ntt.com>, Taku Hasegawa <taku.hasegawa@ntt.com>.

*Proceedings of the 42^{nd} International Conference on Machine Learning*, Vancouver, Canada. PMLR 267, 2025. Copyright 2025 by the author(s).

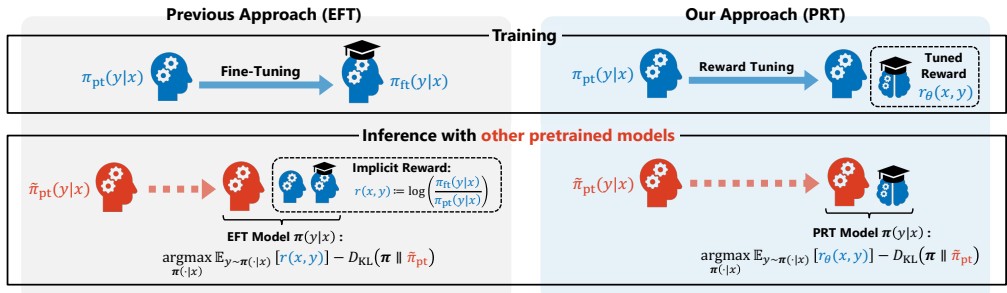

Figure 1: An overview of our approach of portable reward tuning (PRT) compared with the previous work of inference-time tuning, emulated fine-tuning (EFT). In training phase, we tune the reward model $r_\theta(x, y)$ instead of tuning a given pretrained model, through the same loss and dataset, which leads to the reduced cost in inference with another pretrained model.

model in Kullback-Leibler (KL) divergence, whose closed-form solution (Ziebart, 2010) can be computed efficiently and coincides with the fine-tuned model itself in this formulation. Based on this viewpoint, inference-time tuning is performed by solving the reward maximization, with the foundation model in the KL penalty replaced by the new foundation model. This approach is promising since it can avoid the repeated fine-tuning associated with replacement of foundation models, but instead it introduces another issue of inference cost, i.e., requiring three models to run at each inference step.

In this paper, we propose a new principle as an alternative to fine-tuning, called *Portable Reward Tuning* (PRT), that is more suitable for inference-time tuning. The proposed principle is straightforward from the above viewpoint of fine-tuning as reward maximization: we introduce an auxiliary model as an *explicit reward*, and redefine a fine-tuned model, which we call a PRT model, as the closed-form solution that maximizes the explicit reward with KL regularization to a given foundation model. For training, instead of directly optimizing the foundation model itself, the explicit reward model is trained so that the corresponding PRT model minimizes the same cross-entropy loss as in usual fine-tuning. For inference, the PRT model is (re-)constructed from a pair of the explicit reward model and a given (possibly different) foundation model to serve as the corresponding fine-tuned model. By its nature, PRT enables us to freely update the underlying foundation model only with a single model overhead, while the existing methods based on standard fine-tuning still require additional two models to compute the implicit reward for inference-time tuning.

Our contributions are as follows:

- We derive portable reward tuning (PRT) as an alternative to conventional fine-tuning, and established its basic theoretical properties including (i) a natural interpretation of its training objective as reward learning, (ii) evaluation of how the PRT model changes its behavior by replacement of the underlying foundation model in terms of KL divergence, (iii) generalization analysis from the PAC-Bayesian perspective.

- Using both vision and language models, we conduct experiments on inference-time tuning in realistic scenarios like updating the pretrained knowledge and upscaling the size of foundation models. We confirmed that PRT models can achieve comparable accuracy to the baseline of emulated fine-tuning, even with less inference cost in terms of both speed and memory usage.

## 2. Background

In this section, we summarize the background of this work. In Sec. 2.1, we review the basics of KL-regularized Reward Maximization from the literature of Decision-Making Theory and Reinforcement Learning. In Sec. 2.2, we briefly explain previous work on inference-time tuning with its formulation based on reward maximization.

### 2.1. KL-Regularized Reward Maximization

Let $\mathcal{S}$ be a state space, $\mathcal{A}$ be an action space, and a policy model $\pi(a|s)$ be the probability that the action $a \in \mathcal{A}$ is chosen by some probabilistic mapping $\mathcal{S} \rightarrow \mathcal{A}$ given the state $s \in \mathcal{S}$. Furthermore, we consider a reward function $r(s, a) : \mathcal{S} \times \mathcal{A} \rightarrow \mathbb{R}$ and a reference model $\pi_{\text{ref}}(a|s)$. Then the goal of KL-regularized reward maximization is to maximize the expected reward $\mathbb{E}_{a \sim \pi(a|s)}[r(s, a)]$ with a soft constraint that keeps $\pi(a|s)$ close to the reference model $\pi_{\text{ref}}(a|s)$, in the sense of KL divergence, as follows:

$$\max_{\pi(\cdot|s)} \mathbb{E}_{a \sim \pi(a|s)}\left[r\left(s, a\right)\right]$$
$$- \lambda D_{\text{KL}}\left(\pi(\cdot|s) \parallel \pi_{\text{ref}}(\cdot|s)\right), \quad (1)$$

where $D_{\text{KL}}(\pi(\cdot|s) \parallel \pi_{\text{ref}}(\cdot|s)) := \sum_a \pi(a|s) \log(\pi(a|s)/\pi_{\text{ref}}(a|s))$. The closed-form solution of this problem is well-known (Ziebart, 2010; Korbak et al., 2022) as an extension

of maximum entropy principle (Ziebart et al., 2008), given by

$$\pi(a|s) = \frac{1}{Z(s)} \pi_{\text{ref}}(a|s) \exp\left(\frac{1}{\lambda} r(s, a)\right), \qquad (2)$$

where $Z(s) := \sum_{a \in \mathcal{A}} \pi_{\text{ref}}(a|s) \exp\left(\frac{1}{\lambda} r(s, a)\right)$ is the normalization factor.

## 2.2. Inference-Time Tuning

Let $\pi_\theta(y|x)$ be a classification model defined as $\pi_\theta(y|x) :=$ softmax$(f(x; \theta))$, where $x \in \mathcal{X}$ is an input, $y \in \mathcal{Y} = \{c_1, \cdots, c_L\}$ a classification label, $f(x; \theta) \in \mathbb{R}^L$ a neural network parameterized by $\theta \in \mathbb{R}^N$, and the softmax function[1]. Let $\pi_{\text{pt}}(y|x) := \pi_{\theta^{\text{pt}}}(y|x)$ be a pretrained model, and $\pi_{\text{ft}}(y|x) := \pi_{\theta^{\text{ft}}}(y|x)$ its fine-tuned model on some specific task. Typical examples include (i) in image classification, $x$ is an image, $y$ is its corresponding label and $f(x; \theta)$ is a CNN (Bengio et al., 2017) or Vision Transformer (Dosovitskiy et al., 2021); and (ii) in language generation, $x$ is a sequence of tokens $t_1 \cdots t_k$, $y$ is the next token $t_{k+1}$ and $f_\theta(x)$ is a decoder-only Transformer (Radford et al., 2018).

Based on the theory of KL-regularized reward maximization, Mitchell et al. (2024) proposed an emulated fine-tuning (EFT) which views the fine-tuned model $\pi^{\text{ft}}(y|x)$ as the solution of the following problem:

$$\max_{\pi(\cdot|x)} \mathbb{E}_{y \sim \pi(y|x)} \left[ \log \left( \frac{\pi_{\text{ft}}(y|x)}{\pi_{\text{pt}}(y|x)} \right) \right] - D_{\text{KL}}\left(\pi(\cdot|x) \parallel \pi_{\text{pt}}(\cdot|x)\right), \qquad (3)$$

where $\log\left(\pi_{\text{ft}}(y|x)/\pi_{\text{pt}}(y|x)\right)$ is called an *implicit reward*, expected to be the reward function that reflects the task-specific preference for $y \in \mathcal{Y}$. Indeed, by applying (2), the closed-form solution $\pi(y|x)$ of (3) is given by

$$\pi(y|x) = \frac{1}{Z(x)} \pi_{\text{pt}}(y|x) \exp\left( \log\left( \frac{\pi_{\text{ft}}(y|x)}{\pi_{\text{pt}}(y|x)} \right) \right)$$
$$= \pi_{\text{ft}}(y|x)$$

Building on this fact, Mitchell et al. (2024) also proposed what they call scale decoupling, which replaces the pretrained model in the KL constraint by different pretrained model from the one appeared in the implicit reward. In other words, they consider the following problem:

$$\max_{\pi(\cdot|x)} \mathbb{E}_{y \sim \pi(y|x)} \left[ \log \left( \frac{\pi_{\text{ft}}(y|x)}{\pi_{\text{pt}}(y|x)} \right) \right] - D_{\text{KL}}\left(\pi(\cdot|x) \parallel \widetilde{\pi}_{\text{pt}}(\cdot|x)\right), \qquad (4)$$

where $\widetilde{\pi}_{\text{pt}}(\cdot|x)$ is another pretrained model that is different from $\pi_{\text{pt}}(\cdot|x)$, possibly with different network architecture. The closed-form solution of (4) can be considered as the

[1]softmax$(y_1, \cdots, y_L) := (e^{y_1}/\sum_i e^{y_i}, \cdots, e^{y_L}/\sum_i e^{y_i})$

emulation result of fine-tuning the new pretrained model $\widetilde{\pi}_{\text{pt}}(y|x)$ through the implicit reward on the specific task. Also, Liu et al. (2024a) proposed almost the same approach called proxy-tuning.

# 3. Portable Reward Tuning

In this section, we develop a new fine-tuning framework, called portable reward tuning (PRT), with both training and inference algorithms based on KL-regularized reward maximization. Throughout this section, we follow the same classification setting as in Sec. 2.2, which includes both image classification and language generation tasks.

## 3.1. Training of PRT

**Setup.** Let $r(x; \theta) = (r_1(x; \theta), \cdots, r_L(x; \theta)) \in \mathbb{R}^L$ be a neural network with $L$-dimensional outputs. We refer to the $i$-th component of $r(x; \theta)$ as the reward value, denoting $r_\theta(x, c_i) := r_i(x; \theta)$ for an input $x$ and the $i$-th label $c_i \in \mathcal{Y}$. We also assume that a pretrained model $\pi_{\text{pt}}$ and a dataset of input-label pairs $S = \{(x_1, y_1), \cdots, (x_{|S|}, y_{|S|})\}$ for some specific task are given.

**Formulation.** In our PRT framework, we optimize the reward model $r_\theta(x, y)$, instead of directly optimizing the given pretrained model $\pi_{\text{pt}}(y|x)$. For a little while, assume that we already have the learned reward model $r_\theta(x, y)$ for the given specific task. Then, the desired classification model $\pi_\theta(y|x)$ is defined as the solution of reward maximization with KL-constraint to the pretrained model:

$$\max_{\pi(\cdot|x)} \mathbb{E}_{a \sim \pi(y|x)} \left[ r_\theta(x, y) \right] - \lambda D_{\text{KL}}\left(\pi(\cdot|x) \parallel \pi_{\text{pt}}(\cdot|x)\right). \qquad (5)$$

As we already discussed in Sec. 2.1, the closed-form solution for this maximization problem is provided by

$$\pi_\theta(y|x) = \frac{1}{Z_\theta(x)} \pi_{\text{pt}}(y|x) \exp\left( \frac{1}{\lambda} r_\theta(x, y) \right), \qquad (6)$$

where $Z_\theta(x) := \sum_y \pi_{\text{pt}}(y|x) \exp(r_\theta(x, y)/\lambda)$ is the normalization factor. We call this $\pi_\theta(y|x)$ a PRT model (for training) with the reward $r_\theta(x, y)$ and the pretrained model $\pi_{\text{pt}}(y|x)$.

Although it is somewhat obvious, the following proposition guarantees that the above PRT models are equivalent to standard fine-tuned models in terms of their expressiveness:

**Proposition 3.1.** *There is a one-to-one correspondence between fine-tuned models and rewards, which preserves their accuracy:*

$$\{\pi_{\text{ft}}(y|x) : \textit{fine-tuned models}\}$$
$$\rightarrow \{r(y|x) : \textit{rewards satisfying } \mathbb{E}_{y \sim \pi_{\text{pt}}(y|x)}[e^{r(x,y)}] = 1\}$$

*where $\pi_{\mathrm{ft}}(y|x)$ is mapped to the implicit reward* $\log(\pi_{\mathrm{ft}}(y|x)/\pi_{\mathrm{pt}}(y|x))$.

*Proof.* The mapping preserves accuracy since the PRT model (6) with the implicit reward recovers the given model $\pi_{\mathrm{ft}}(y|x)$. The invertibility also holds since the implicit reward of the PRT model (6) recovers the reward itself. $\square$

In other words, if there is a fine-tuned model that achieves some accuracy, there is also the corresponding reward whose PRT model achieves the same accuracy. Thus, the reparameterization in PRT (6) does not restrict its expressiveness, even compared to standard fine-tuning.

**Training objective.** The reward model $r_\theta(x, y)$ is trained by simply optimizing the same loss function $\mathcal{L}(\mathbf{p}, y^*)$ as in standard fine-tuning, with the true label $y^*$ for the input $x$ and the output distribution $\mathbf{p} := \pi_\theta(\cdot|x)$ of the PRT model. (See also lines 6-9 in Algorithm 1.) In particular, throughout this paper, we minimize the cross-entropy loss $\mathcal{L}(\mathbf{p}, y^*) := \mathrm{CE}(\mathbf{p}, y^*) := -\log \pi_\theta(y^*|x)$ over a given dataset $S \subset \mathcal{X} \times \mathcal{Y}$ to train the reward model:

$$\arg\min_\theta \frac{1}{|S|} \sum_{(x,y^*) \in S} \mathcal{L}(\mathbf{p}, y^*) \qquad (7)$$

$$= \arg\max_\theta \sum_{(x,y^*) \in S} \log \pi_\theta(y^*|x)$$

$$= \arg\max_\theta \sum_{(x,y^*) \in S} r_\theta(x, y^*) - V_\theta(x), \qquad (8)$$

where $V_\theta(x) := \lambda \log Z_\theta(x)$. Notably, this maximization can be reinterpreted in terms of reward training. Indeed, by applying Jensen's inequality, we obtain

$$r_\theta(x, y^*) - V_\theta(x)$$
$$= r_\theta(x, y^*) - \lambda \log \mathbb{E}_{y \sim \pi_{\mathrm{pt}}(y|x)} \exp(r_\theta(y|x)/\lambda)$$
$$\leq r_\theta(x, y^*) - \lambda \mathbb{E}_{y \sim \pi_{\mathrm{pt}}(y|x)} \log \exp(r_\theta(y|x)/\lambda)$$
$$= r_\theta(x, y^*) - \mathbb{E}_{y \sim \pi_{\mathrm{pt}}(y|x)} r_\theta(y|x). \qquad (9)$$

Therefore, *maximization of (8), i.e., training the reward model using the cross-entropy loss, leads to an increase in the reward for the ground-truth $y^*$ while decreasing rewards for the average outcomes $y$ from the pretrained model $\pi_{\mathrm{pt}}(y|x)$.* This interpretation can be seen as analogous to the training of the Bradley-Terry reward model in RLHF (Christiano et al., 2017) with pairs of preferred-dispreferred sentences, where the reward model learns to evaluate the preferred one higher than the dispreferred one.

**Implementation Details** Algorithm 1 presents the pseudocode for training PRT models. The training data is a set of input-label pairs just like in standard training. For the

---

**Algorithm 1** Pseudocode for Training of PRT

1: **Given:** training data $S = \{(x_1, y_1), \cdots, (x_m, y_m)\}$,
2:     a reward model $r(x; \theta)$,
3:     a pretrained model $\pi_{\mathrm{pt}}(\cdot|x) = \mathrm{softmax}(f(x; \theta_{\mathrm{pt}}))$.
4: Initialize $\theta$.
5: **for** $i = 1, ..., m$ **do**
6:     $\mathbf{v}_\theta \leftarrow \log \mathrm{softmax}(f(x_i; \theta_{\mathrm{pt}})) + r(x_i; \theta)$.
7:     $\mathbf{p}_\theta \leftarrow \mathrm{softmax}(\mathbf{v}_\theta)$
8:     $\mathcal{L}_\theta \leftarrow \mathrm{CE}(\mathbf{p}_\theta, y_i)$
9:     Update $\theta$ with the gradient of $\mathcal{L}_\theta$.
10: **end for**
11: **return** $\theta$.

---

**Algorithm 2** Pseudocode for Inference of PRT

1: **Given:** an input $x$, the trained reward $r(x; \theta)$,
2:     a pretrained model $\widetilde{\pi}_{\mathrm{pt}}(y|x) = \mathrm{softmax}(\widetilde{f}(x; \widetilde{\theta}_{\mathrm{pt}}))$,
3: $\widetilde{\mathbf{v}} \leftarrow \log \mathrm{softmax}(\widetilde{f}(x; \widetilde{\theta}_{\mathrm{pt}})) + r(x; \theta)$.
4: $\widetilde{\mathbf{p}} \leftarrow \mathrm{softmax}(\widetilde{\mathbf{v}})$
5: **return** $\widetilde{\mathbf{p}}$ as the output probability conditioned by $x$.

---

reward model $r(x; \theta)$, although it can be an arbitrary neural network model, we assume that it is modeled using the same network architecture as the pretrained model $f(x; \theta_{\mathrm{pt}})$ and that $\theta$ initialized with $\theta_{\mathrm{pt}}$ throughout this paper. Lines 6-7 compute the PRT model (6) in the logit space to avoid numerical instability. Here, we set the coefficient $\lambda = 1$ in (6) since the reward model can automatically learn the scaling factor.

## 3.2. Inference of PRT

Let $r_\theta(x, y)$ and $\pi_{\mathrm{pt}}(y|x)$ be the reward and pretrained model introduced in Section 3.1. Let $\widetilde{\pi}_{\mathrm{pt}}(y|x)$ be another pretrained model whose label space $\mathcal{Y}$ (or vocabularies for language models) is the same as the one for $\pi_{\mathrm{pt}}(y|x)$. Examples of $\widetilde{\pi}_{\mathrm{pt}}(y|x)$ include a model pretrained on a larger or more recent dataset than that of $\pi_{\mathrm{pt}}(y|x)$, and a pretrained model with more parameters.

The inference model $\widetilde{\pi}_\theta(y|x)$ for the reward $r_\theta(x, y)$ and the specified pretrained model $\widetilde{\pi}_{\mathrm{pt}}(y|x)$ can be derived by replacing $\pi_{\mathrm{pt}}(y|x)$ in (6) with $\widetilde{\pi}_{\mathrm{pt}}(y|x)$. Specifically, given an input $x \in \mathcal{X}$, the prediction for its label $y$ is performed by the following model that maximizes the reward $r_\theta(x, y)$ while minimizing the deviation from the specified pretrained model $\widetilde{\pi}_{\mathrm{pt}}(y|x)$:

$$\widetilde{\pi}_\theta(y|x) := \arg\max_{\pi(\cdot|x)} \mathbb{E}_{a \sim \pi(y|x)} [r_\theta(x, y)]$$
$$- \lambda D_{\mathrm{KL}} (\pi(\cdot|x) \| \widetilde{\pi}_{\mathrm{pt}}(\cdot|x))$$
$$= \frac{1}{\widetilde{Z}_\theta(x)} \widetilde{\pi}_{\mathrm{pt}}(y|x) \exp \left( \frac{1}{\lambda} r_\theta(x, y) \right), \qquad (10)$$

where $\widetilde{Z}_\theta(x) := \sum_y \widetilde{\pi}_{\mathrm{pt}}(y|x) \exp(r_\theta(x, y)/\lambda)$. The implementation of inference by this PRT model (10) is straightforward as described in Algorithm 2, with $\lambda = 1$ as in training.

Now the following question naturally arises: How does the choice of $\widetilde{\pi}_{\mathrm{pt}}(y|x)$ affect the behavior of the inference model $\widetilde{\pi}(y|x)$? Intuitively, if $\widetilde{\pi}_{\mathrm{pt}}(y|x)$ does not deviate from the original $\pi_{\mathrm{pt}}(y|x)$, the inference model $\widetilde{\pi}_{\mathrm{pt}}(y|x)$ also keeps to behave similarly to the training time. This intuition can be formalized as follows:

**Proposition 3.2.** *Suppose that $\widetilde{\pi}_{\mathrm{pt}}(y|x)$ is close to $\pi_{\mathrm{pt}}(y|x)$, i.e., $D_{\mathrm{KL}}\left(\pi_{\mathrm{pt}}(\cdot|x) \,\|\, \widetilde{\pi}_{\mathrm{pt}}(\cdot|x)\right) \leq \varepsilon$. Additionally, we assume that the maximum and mean value ratio of the exponential reward, i.e., $\max_y \exp r_\theta(x,y) / \mathbb{E}_y \exp r_\theta(x,y)$, is bounded by some constant $C$. Then, the PRT models $\pi_\theta(y|x)$ and $\widetilde{\pi}_\theta(y|x)$ are also close as distributions:*

$$D_{\mathrm{KL}}\left(\pi_\theta(y|x) \,\|\, \widetilde{\pi}_\theta(y|x)\right) \leq O(\sqrt{\varepsilon}).$$

*Proof.* See Appendix A. $\square$

### 3.3. A PAC-Bayesian Perspective

We can suppose that the pretrained models for both training and inference, i.e., $\pi_{\mathrm{pt}}(y|x)$ and $\widetilde{\pi}_{\mathrm{pt}}(y|x)$, are chosen from some distribution $\mathcal{P}$ over the set of pretrained models. Then the PRT models for inference, $\widetilde{\pi}_\theta(y|x)$ combined with the sampled pretrained model $\widetilde{\pi}_{\mathrm{pt}}(y|x) \sim \mathcal{P}$, form a new distribution $\mathcal{Q}_\theta$.

This formulation of PRT models naturally fits into the PAC-Bayes framework established in McAllester (1999), which enables us to analyze the generalization error of posterior distributions over classifiers, in comparison to a fixed prior distribution. Specifically in our setting, the pretrained distribution $\mathcal{P}$ can be seen as a prior distribution, and the PRT distribution $\mathcal{Q}_\theta$ as a posterior distribution.

Let $l(x, \pi)$ be a finitely bounded loss function, e.g., one that returns the error rate, for a given input $x$ and classifier $\pi$. Assume that the input $x$ follows some distribution $\mathcal{D}$. The following generalization bound can be obtained as a direct consequence of Theorem 1 in McAllester (1999):

**Proposition 3.3.** *Let $S = (x_1, \cdots, x_m) \sim \mathcal{D}^m$ be i.i.d. $m$ training samples from the data distribution $\mathcal{D}$. Then, with probability at least $1 - \delta$, we have*

$$\mathbb{E}_{\pi \sim \mathcal{Q}_\theta} \mathbb{E}_{x \sim \mathcal{D}}\left[l(x, \pi)\right] \leq \mathbb{E}_{\pi \sim \mathcal{Q}_\theta}\left[\frac{1}{m}\sum_i^m l(x_i, \pi)\right]$$
$$+ \sqrt{\frac{D_{\mathrm{KL}}(\mathcal{Q}_\theta \,\|\, \mathcal{P}) + \log(\frac{1}{\delta}) + \frac{5}{2}\log m + 8}{2m - 1}}, \quad (11)$$

*for the posterior distribution $\mathcal{Q}_\theta$ with the reward $r_\theta$ trained on $S$.*

The KL divergence $D_{\mathrm{KL}}(\mathcal{Q}_\theta \,\|\, \mathcal{P})$ is not computationally tractable because the pretrained distribution $\mathcal{P}$ itself is not tractable and also the underlying space of pretrained models is too vast. Nevertheless, the result implies that the generalization capability of PRT models can be captured by the closeness of the PRT model $\widetilde{\pi}_\theta(y|x)$ compared to the underlying pretrained model $\widetilde{\pi}_{\mathrm{pt}}(y|x)$.

In particular, we can easily see that the KL divergence term vanishes if the reward $r_\theta(x, y)$ is a constant value given each $x$ and then the equality $\widetilde{\pi}_\theta(y|x) = \widetilde{\pi}_{\mathrm{pt}}(y|x)$ holds. This can be seen as the case that the exponential distribution $\rho_\theta(y|x) := \exp(r_\theta(x, y))/\sum_y \exp(r_\theta(x, y))$ maximizes its entropy. Thus, the generalization capability can be further improved by regularizing the exponential distribution of the reward $r_\theta(x, y)$ during training of $r_\theta(x, y)$, while it may hurt the optimization quality instead.

The above analysis motivates us to consider the following regularization, which we call Entropy Maximization (EM), to optionally enhance the generalization capability across various pretrained models:

$$\widetilde{\mathcal{L}}(\theta) := \mathcal{L}(\theta) - \alpha \frac{1}{|S|} \sum_{(x,y) \in S} H(\rho_\theta(\cdot|x)), \quad (12)$$

where $\mathcal{L}(\theta)$ is the original loss (7) for training the reward model, $H(\rho_\theta(\cdot|x)) := -\sum_{y \in \mathcal{Y}} \rho_\theta(y|x) \log \rho_\theta(y|x)$ is the entropy of the exponential distribution $\rho_\theta(y|x)$, and $\alpha \in \mathbb{R}_{\geq 0}$ is the hyperparameter to control the regularization. As we can see larger $\alpha$ enforces the reward $r_\theta(x, y)$ to be closer to some constant for each input $x \in \mathcal{X}$.

## 4. Experiments

In this section, we evaluate the performance of portable reward tuning (PRT) for inference-time tuning, with various pretrained models including both vision and language models. The main comparison is between PRT and its corresponding baseline, emulated fine-tuning (EFT; Mitchell et al. (2024)), which differ only in whether they maximize an explicit or implicit reward during inference-time tuning.

**Setups for vision experiments.** (1) Pretrained models: We employed CLIP models (Ilharco et al., 2021) pretrained on various datasets including OpenAI's proprietary dataset (Radford et al., 2021), LAION-400M (Schuhmann et al., 2021), LAION-2B (Schuhmann et al., 2022), and DataComp-1B (Gadre et al., 2024). (2) Fine-tuning: For each fine-grained dataset, such as Cars (Krause et al., 2013) and CUB (Wah et al., 2011), we first constructed and fixed the classification layer of each pretrained model for zero-shot classification, and then fine-tuned (or reward-tuned) its feature extractor on the train set. (3) Evaluation: We evaluated models on the test set of each dataset where the training set was used for tuning the models.

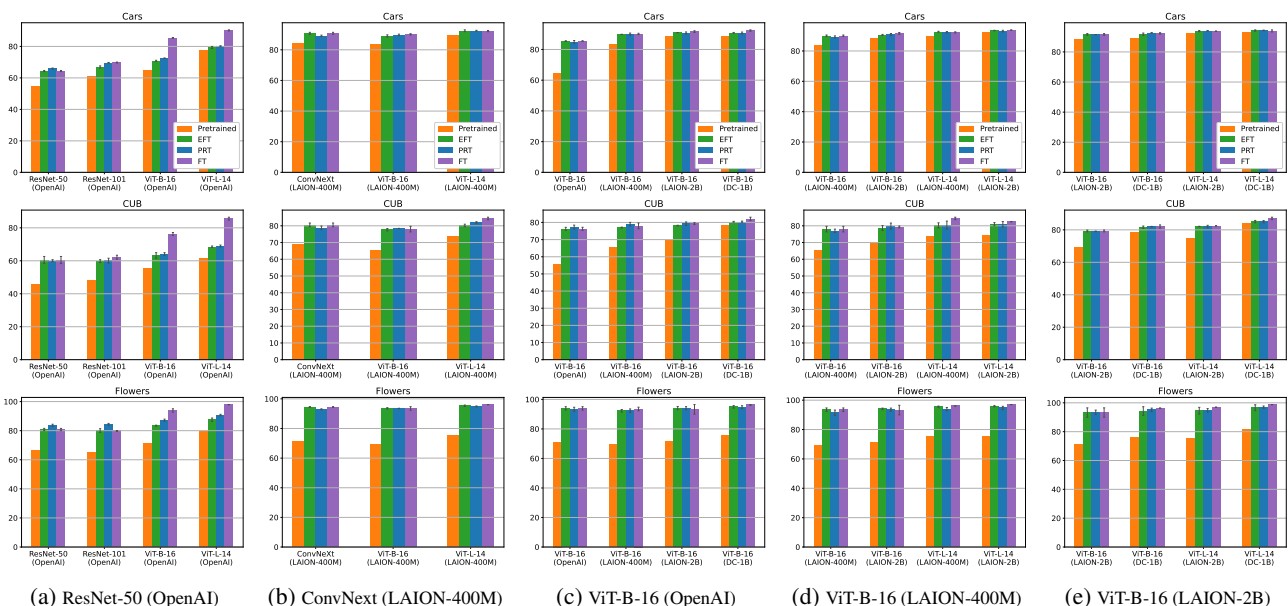

Figure 2: Evaluations of inference-time tuned models for vision tasks. Each subcaption refers to the source pretrained model, and the labels in x-axis are target pretrained models. **Pretrained** means the zero-shot classification by each target model as a baseline, and **FT** means the fine-tuned target model as an oracle result.

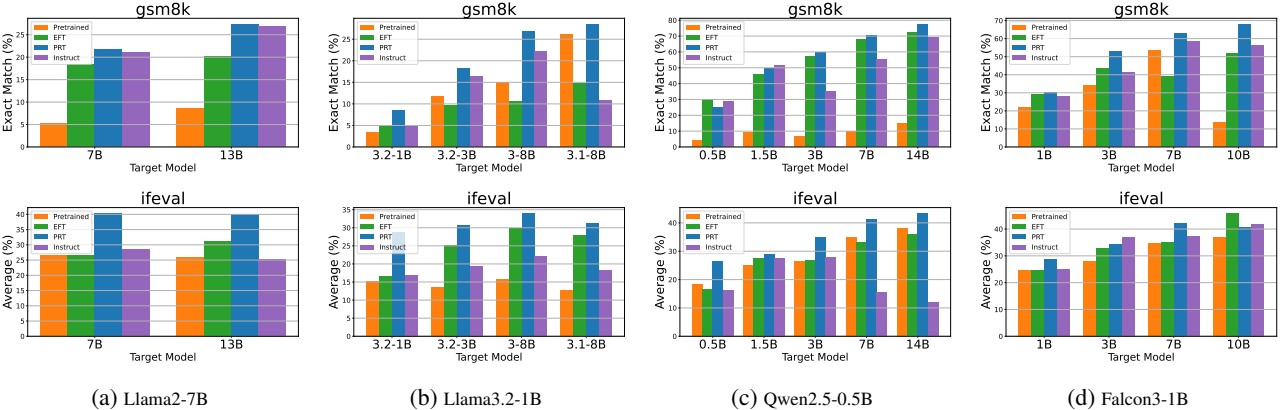

Figure 3: Evaluations of inference-time instruction-tuned models on GSM8k and IFEval benchmarks. Each subcaption refers to the source pretrained model, and the labels in x-axis are target pretrained models. **Pretrained** means the zero-shot inference by each target model as a baseline, and **Instruct** means the instruct-tuned target model as an oracle result.

**Setups for language experiments.** (1) Pretrained models: We employed pretrained language models of the decoder-only Transformers such as LLAMA series (Touvron et al., 2023b), Qwen series (Yang et al., 2024b;a) and Falcon series (Team, 2024). (2) Fine-tuning: We performed instruction tuning on these pretrained models with Tulu v2 dataset (Ivison et al., 2023), a large-scale dataset consisting of demonstrations for following given instructions. (3) Evaluation: We evaluated models on downstream benchmarks, particularly on the GSM8k benchmark (Cobbe et al., 2021) for reasoning ability and IFEval benchmark (Zhou et al., 2023) for instruction-following ability.

### 4.1. Results

Figures 2, 3 and 4 show the results of inference-time tuning using PRT and EFT, from a *source* pretrained model, i.e., the one used in tuning the (either explicit or implicit) reward, to a *target* pretrained model, i.e., the one never used in tuning the reward. Here PRT refers to the vanilla one without regularization for fair comparison. We also compare them with the zero-shot performance of pretrained models themselves as baselines, and with their fine-tuned performance as oracles. Note that, in the case that the source and target are the same pretrained model (i.e., the leftmost one in x-axis), the results correspond to the ones without inference-time

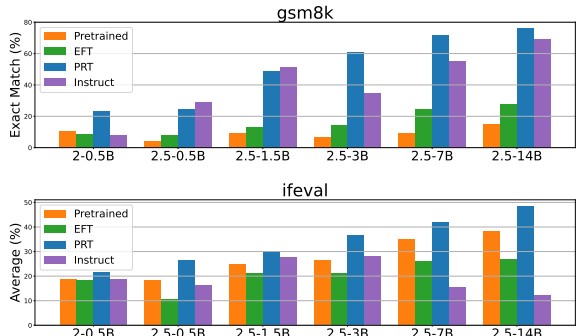

Figure 4: Inference-time tuning from Qwen2-0.5B to the Qwen2.5 models with various sizes.

tuning. Overall, these results support our main claim in this paper, i.e., PRT achieves comparable accuracy to EFT with less inference cost. (See also Section 4.4)

Particularly, the pairs of the source/target pretrained models can be categorized into either of the following two scenarios: (i) upscaling the source model to larger target models, and (ii) updating the pretrained knowledge of the source model to the target model with better pretraining data. The results for the former scenario (i) are shown in Figures 2a-2b, 3a-3d, where the source and target models are pretrained on the same dataset, and thus the only difference between them is the network architecture. The results for the latter scenario (ii) are shown in Figures 2c-2e and Figure 4. In both scenarios, we observe that PRT successfully leverages the improved capabilities of target pretrained models, by reusing the fixed reward model trained with the source pretrained model.

## 4.2. Qualitative Analysis

To analyze the behavior of PRT in more detail, we examine the tokens generated by the model. In this analysis, we used Llama3-8B as the target model, while Llama-3.2-1B was used for training and reward-model initialization. Figure 5 presents the output obtained under these settings. We observe that the PRT output correctly produces the chain of reasoning that leads to the final answer of 21.[2] Additionally, Figures 5a and 5b show the top-5 prediction probabilities for the tokens that follow each highlighted sentence. From these distributions, we see that the target pretrained model's predictions are altered by the PRT model. In Figure 5b, for instance, the pretrained model attempts to output the answer "33" directly, whereas the PRT outputs "6" which reflects the step-by-step reasoning capability acquired through inference-time instruction-tuning.

---

[2]By contrast, the pretrained Llama3-8B output was only "18." For details, see Appendix D.

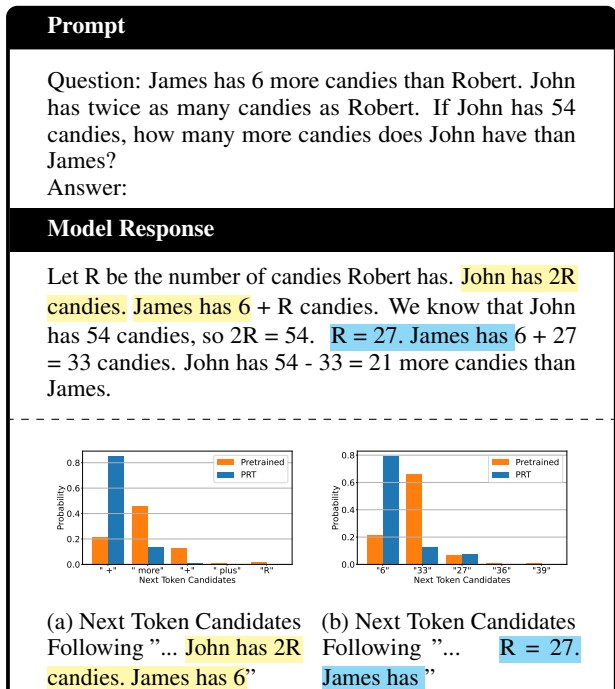

Figure 5: Changes in next-token probabilities by PRT.

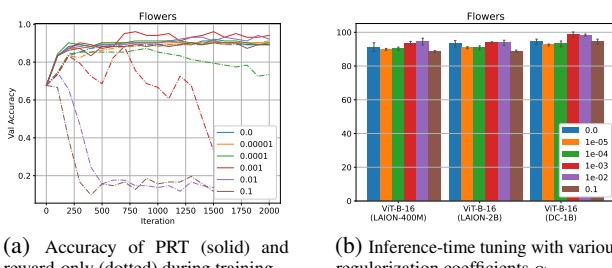

(a) Accuracy of PRT (solid) and reward-only (dotted) during training.

(b) Inference-time tuning with various regularization coefficients $\alpha$.

Figure 6: Analysis on how the EM regularization affects on performance of PRT with various coefficient $\alpha$. Reward-only refers to the exponential distribution $\rho_\theta(y|x)$ in Sec 3.3.

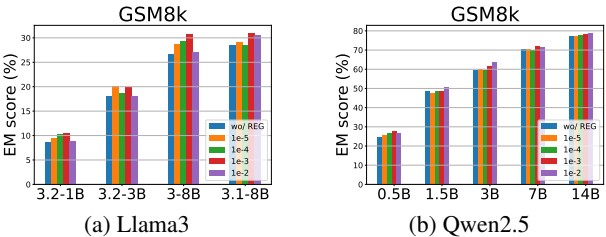

(a) Llama3

(b) Qwen2.5

Figure 7: Performance of PRT on GSM8k with various regularization coefficients $\alpha$.

## 4.3. Effects of Entropy Regularization

Here we employ Entropy Maximization (EM) regularization in PRT training, with its coefficient $\alpha \in \mathbb{R}_{\geq 0}$. In Section 3.3, the PAC-Bayesian analysis indicated that the generalization capability of PRT models is affected by the entropy of the reward distribution $\rho_\theta(y|x)$ defined by the reward model

$r_\theta(x, y)$. In Figure 6 and 7, we empirically analyze its effect by varying $\alpha$. First of all, we observed that the EM regularization generally suppresses the accuracy of the reward itself (Fig. 6a) as $\alpha$ increases, which can be naturally expected. However, interestingly, the performance of PRT (Fig. 6b, 7) is not degraded even with such rewards, but also sometimes boosted regardless of the accuracy of rewards themselves. The regularization itself may not be yet practical since the optimal $\alpha$ tends to be dependent on tasks and pretrained models, but these analyses provide valuable insights for generalization capability of PRT models, which may lead to future exploration of more practical regularization.

## 4.4. Memory and Speed Analysis

Since PRT introduces an auxiliary model as the reward for both training and inference, we investigate how the memory usage and speed increase or decrease, compared to standard fine-tuning in training and EFT in inference. For training, Table 1 in Appendix shows that, while training time slightly increases due to the auxiliary model, the increase in memory usage is relatively negligible because the pretrained model in PRT does not require back-propagation. For inference, Tables 2 and 3 show that PRT successfully reduces both inference time and memory usage compared to EFT, which highlights the benefit of employing explicit reward models.

## 5. Related Work

**Tuning by Refining Predictions.** In this paper we focused on inference-time tuning accomplished by refining predictions from the underlying pretrained model. In particular, our baseline is the emulated fine-tuning (EFT; Mitchell et al. (2024)) which established the interpretation of inference-time tuning based on KL-regularized reward maximization. Parallel work by Liu et al. (2024a) also proposed an essentially same method called proxy-tuning. While these previous work focused on inference-time tuning with pretrained models that only differ in their model scale, we examined a more general setting that the pretrained models may differ even in their architectures or pretraining datasets. Also, the previous work assumed the fine-tuned model was prepared in advance, and explored how to exploit it for a new pretrained model. In contrast, we reexamined the assumption and explored an alternative to fine-tuning that is more suitable for inference-time tuning. As a consequence, at the cost of a little overhead in training, our approach successfully halves the overhead in inference-time tuning.

The literature of controlled text generation (Krause et al., 2021; Yang & Klein, 2021; Pascual et al., 2021; Li et al., 2023; Deng & Raffel, 2023) is also related to our work, but they have explored specific methods for attribute-conditioned text generation, which requires a classifier for some attributes, rather than fine-tuning for general tasks as

in our work. In particular, Deng & Raffel (2023) proposed to control language models with reward models similarly to our work, but which are trained with a manually-designed loss specific to text classification tasks, while our method employs standard loss for fine-tuning and thus can be naturally applied to broader domains and tasks such as vision classification and instruction tuning.

**Tuning by Editing Parameters or Activations.** Another possible approach for inference-time tuning would be directly editing the parameters or activations of pretrained models, instead of their predictions. A bunch of research on parameter editing, including (Ilharco et al., 2023; Gueta et al., 2023; Ortiz-Jimenez et al., 2024; Yadav et al., 2024; Chijiwa, 2024; Daheim et al., 2024), addresses inference-time tuning by leveraging existing fine-tuned results. However, most of these work aimed to tune a given pretrained model for multiple tasks, and thus cannot be used to tune a newly provided pretrained model. Although Chijiwa (2024) tackled the challenge of inference-time tuning with different pretrained models, it still requires the pretrained models to share the same architecture, which may be a fundamental limitation of the approach by editing parameters. Similarly, there is a line of research (Dathathri et al., 2020; Hernandez et al., 2023; Chuang et al., 2024; Li et al., 2024) on tuning by editing the activations, but they also require the model architecture to have the same dimension for activations. In contrast to these approaches, the approach by refining predictions in this paper would be more promising since it is completely free from the choice of the model architectures. Also, in cases of language models, automatic prompt tuning (Zhou et al., 2022; Pryzant et al., 2023; Wang et al., 2024b) can also be considered as an approach of tuning models by editing input prompts and reusable to other language models, though there still have been difficulties coming from both the discrete nature of prompts and their limited expressiveness.

**Reward for Language Models.** For language models, the notion of rewards has been exploited mainly in two lines of research: (i) Reinforcement Learning from Human Feedback (RLHF; Christiano et al. (2017); Jaques et al. (2017); Ouyang et al. (2022)), and (ii) multi-step reasoning (Cobbe et al., 2021; Uesato et al., 2022). In RLHF, reward models are first trained on a dataset of human feedbacks, such as pairs of prefered-disprefered responses, and then used for training LLMs by reinforcement learning. Although recent methods (Rafailov et al., 2024a; Calandriello et al., 2024; Ethayarajh et al., 2024; Rafailov et al., 2024b) successfully bypass the explicit use of reward models, they are still learning human preferences through implicit rewards. Several work (Liu et al., 2024b; Chakraborty et al., 2024; Khanov et al., 2024) also integrate the idea of inference-time decoding into RLHF, but they assume the reward model is already prepared by RLHF. In multi-step reasoning, reward

models are trained to evaluate the intermediate process of reasoning by LLMs, from datasets with process or outcome supervision, and then used as verifiers for sampling such as Best-of-N sampling (Lightman et al., 2024; Wang et al., 2024a) or self-consistency decoding (Wang et al., 2023; Luo et al., 2024). In contrast to these lines of work, where rewards are used for either training LLMs or verifying inference, the reward model in our approach is directly trained for *classification or generation* from ground-truth labels or tokens through the same loss and dataset for standard fine-tuning, which may lead to a broad range of applications in various domains including language generation.

## 6. Conclusion

In this paper, we introduced a new fine-tuning principle called portable reward tuning (PRT) as an alternative to standard fine-tuning, based on the interpretation of fine-tuning as reward maximization with KL regularization. PRT naturally fits into the framework of inference-time tuning as the reward maximization. We theoretically analyzed its basic properties of both training and inference, revealing how the reward evolves in training and how the choice of pretrained models affects inference-time tuning. Also we empirically confirmed that PRT can achieve comparable accuracy to previous work of inference-time tuning, even with less computational overhead.

## Impact Statement

This paper investigates basic properties of a proposed principle of fine-tuning, which aims to advance the field of Machine Learning. There may be potential societal consequences of our work, but discussing about them is out of scope of this work.

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

# A. Proof of Proposition 3.2

*Proof.* Let $Z_\theta(x) := \mathbb{E}_{y \sim \pi_{\mathrm{pt}}(y|x)}[\exp(r_\theta(x,y))]$ and similarly $\widetilde{Z}_\theta(x) := \mathbb{E}_{y \sim \widetilde{\pi}_{\mathrm{pt}}(y|x)}[\exp(r_\theta(x,y))]$ for convenience.

$$
D_{\mathrm{KL}}(\pi_\theta(\cdot|x) \| \widetilde{\pi}_\theta(\cdot|x))
$$

$$
= \sum_y \pi_\theta(y|x) \log \left( \frac{\pi_\theta(y|x)}{\widetilde{\pi}_\theta(y|x)} \right)
$$

$$
= \sum_y \frac{\pi_{\mathrm{pt}}(y|x) \exp(r_\theta(x,y))}{Z_\theta(x)} \log \left( \frac{\pi_\theta(y|x)}{\widetilde{\pi}_\theta(y|x)} \right)
$$

$$
= \sum_y \frac{\pi_{\mathrm{pt}}(y|x) \exp(r_\theta(x,y))}{Z_\theta(x)}
$$

$$
\times \left\{ \log \left( \frac{\pi_{\mathrm{pt}}(y|x)}{\widetilde{\pi}_{\mathrm{pt}}(y|x)} \right) + \log \left( \frac{\widetilde{Z}_\theta(x)}{Z_\theta(x)} \right) \right\}
$$

$$
= \frac{1}{Z_\theta(x)} \sum_y \exp(r_\theta(x,y)) \pi_{\mathrm{pt}}(y|x) \log \left( \frac{\pi_{\mathrm{pt}}(y|x)}{\widetilde{\pi}_{\mathrm{pt}}(y|x)} \right)
$$

$$
+ \log \left( \frac{\widetilde{Z}_\theta(x)}{Z_\theta(x)} \right)
$$

$$
\leq \frac{\max_y \exp r_\theta(x,y)}{\mathbb{E}_{y \sim \pi_{\mathrm{pt}}(y|x)} \exp r_\theta(x,y)} D_{\mathrm{KL}}(\pi_{\mathrm{pt}}(\cdot|x) \| \widetilde{\pi}_{\mathrm{pt}}(\cdot|x))
$$

$$
+ \log \left( \frac{\widetilde{Z}_\theta(x)}{Z_\theta(x)} \right).
$$

The first term can be bounded by $C\varepsilon$ by combining the assumptions. The second term can be evaluated as follows:

$$
\frac{\widetilde{Z}_\theta(x)}{Z_\theta(x)}
$$

$$
= \frac{\sum_y \widetilde{\pi}_{\mathrm{pt}}(y|x) \exp r_\theta(x,y)}{\sum_y \pi_{\mathrm{pt}}(y|x) \exp r_\theta(x,y)}
$$

$$
= 1 + \frac{\sum_y (\widetilde{\pi}_{\mathrm{pt}}(y|x) - \pi_{\mathrm{pt}}(y|x)) \exp r_\theta(x,y)}{\sum_y \pi_{\mathrm{pt}}(y|x) \exp r_\theta(x,y)}
$$

$$
\leq 1 + \frac{\max_y \exp r_\theta(x,y)}{\mathbb{E}_{y \sim \pi_{\mathrm{pt}}(y|x)} \exp r_\theta(x,y)} \sum_y \left| \widetilde{\pi}_{\mathrm{pt}}(y|x) - \pi_{\mathrm{pt}}(y|x) \right|
$$

$$
\leq 1 + C \sum_y \left| \widetilde{\pi}_{\mathrm{pt}}(y|x) - \pi_{\mathrm{pt}}(y|x) \right|
$$

(by the assumption)

$$
\leq 1 + C' \sqrt{D_{\mathrm{KL}}(\widetilde{\pi}_{\mathrm{pt}}(\cdot|x) \| \pi_{\mathrm{pt}}(\cdot|x))}
$$

(by Pinsker's inequality)

$$
\leq 1 + C' \sqrt{\varepsilon}.
$$

Since $\log(1 + C'\sqrt{\varepsilon}) = O(\sqrt{\varepsilon})$ holds asymptotically, finally we have $D_{\mathrm{KL}}(\pi_\theta(y|x) \| \widetilde{\pi}_\theta(y|x))) \leq O(\sqrt{\varepsilon})$. $\qquad \square$

# B. Experimental Setup

## B.1. Image Classification Tasks.

**Training Setups**    In training of either standard fine-tuning or PRT, we used the same hyperparameters following existing work (Ilharco et al., 2023) as follows: learning rate $= 1 \times 10^{-5}$, batch size $= 128$, number of iterations $= 2000$, optimizer $=$ Adam, cosine annealing with $500$ warmup iterations. All models are trained on a single A100 GPU.

**Fine-Tuning and Evaluation Dataset**    We consider the following image classification tasks:

- **Aircraft** (Maji et al., 2013): A dataset with 100 classes of aircrafts, 100 images per class.

- **Caltech101** (Li et al., 2022): A dataset with 101 classes of objects, 40 to 800 images per class.

- **Cars** (Krause et al., 2013): A dataset with 196 classes of various cars.

- **CIFAR-100** (Krizhevsky et al., 2009): A dataset with 100 classes of 32x32 color images.

- **Country211** (Radford et al., 2021): A dataset of photos taken in 211 different countries.

- **CUB** (Wah et al., 2011): A dataset of images with 200 bird species for fine-grained classification.

- **Flowers** (Nilsback & Zisserman, 2008): The Oxford 102 Flower Dataset, containing images of 102 flower categories for fine-grained classification.

- **RESISC45** (Cheng et al., 2017): A dataset of images with 45 scene classes.

**Models**    We employed the following model architectures for the vision feature extractors of CLIP models:

- **ResNets (ResNet-50, ResNet-101; He et al. (2016))**

- **ConvNext (Liu et al., 2022)**

- **Vision Transformers (ViT-B-16, ViT-L-14; Dosovitskiy et al. (2021))**

## B.2. Language Modeling Tasks.

**Training Dataset and Settings**    For instruction tuning, we used the Tulu v2 dataset (Ivison et al., 2023), which is a large-scale dataset designed to improve the instruction-following capabilities of language models. The dataset includes a diverse set of instructions and corresponding responses, covering a wide range of topics and tasks. The training conditions are as follows: learning rate $= 2 \times 10^{-5}$, batch size $= 128$, number of epochs $= 2$, optimizer $=$ Adam, warmup ratio $= 0.03$, and learning rate scheduler $=$ linear. We conducted all training on 8 NVIDIA A100 GPUs.

**Evaluation Dataset**    We consider the following language modeling tasks: GSM8k and IFEval.

- **GSM8K** (Cobbe et al., 2021): A dataset for evaluating the ability of models to solve grade-school math problems. Evaluation is based on the exact match metric, which measures whether the model's final answer exactly matches the correct answer.

- **IFEval** (Zhou et al., 2023): A dataset for evaluating the ability of models to perform information extraction tasks. It employs four evaluation metrics: instruction-level strict accuracy, instruction-level relaxed accuracy, prompt-level strict accuracy, and prompt-level relaxed accuracy, which comprehensively assess the model's ability to follow complex instructions. In this paper, we report the average of these four scores as the evaluation metric.

**Models**    We evaluated the proposed PRT method on the following models:

- **LLAMA 2 Series** ([Touvron et al., 2023a](#)): A family of large-scale language models developed by Meta AI, available in various sizes, including 7B, 13B, and 70B parameters, designed for a wide range of natural language processing tasks.

- **LLAMA 3 Series** ([Dubey et al., 2024](#)): A family of large-scale language models developed by Meta AI, introduced in April 2024 with 8B and 70B parameter variants. Compared to Llama 2, it features improvements in tokenizer efficiency, training data scale, and overall model optimization. The subsequent update, **LLAMA 3.1**, released in July 2024, further enhanced performance by refining pretraining methodologies while maintaining the same parameter sizes. In September 2024, **LLAMA 3.2** introduced lightweight text models with 1B and 3B parameters, optimized for efficiency in resource-constrained environments such as mobile and edge devices.

- **Qwen 2 Series** ([Yang et al., 2024b;a](#)): A series of large-scale language models developed by Alibaba Cloud's Qwen team, designed for various natural language understanding and generation tasks. The initial **Qwen 2** models were released with parameter sizes such as 1.5B and 3B, focusing on high-quality training data and diverse applications. The subsequent **Qwen 2.5** series expanded the model range, introducing sizes from 0.5B to 72B parameters, with both base and instruction-tuned variants, further improving performance and efficiency.

- **Falcon 3 Series** ([Team, 2024](#)): A series of open-source large language models developed by the Technology Innovation Institute (TII) in Abu Dhabi, designed to provide accessible and efficient AI solutions. Released in December 2024, Falcon 3 models are available in 1B, 3B, 7B, and 10B parameter sizes, each offered in both Base and Instruct variants. The Base models are tailored for general-purpose text generation, while the Instruct models are fine-tuned for conversational applications.

## C. Memory and Speed Benchmarks

See Table 1 for training-time benchmarks, and Tables 2 and 3 for inference-time benchmarks.

| Models | | FT | PRT |
|---|---|---|---|
| ResNet-50 | Peak GPU memory | 12.84 GB | 13.08 GB |
| | Average time per batch | $38.26_{\pm 3.62}$ ms | $46.36_{\pm 3.34}$ ms |
| ViT-B-16 | Peak GPU memory | 20.17 GB | 20.58 GB |
| | Average time per batch | $116.10_{\pm 0.43}$ ms | $151.13_{\pm 0.48}$ ms |

Table 1: Memory usage and average time per batch in training with batch size 128.

| Source Models | Target Models | | Target FT (Oracle) | EFT | PRT |
|---|---|---|---|---|---|
| ResNet-50 | ViT-B-16 | Peak GPU memory | 1.46 GB | 2.42 GB | 2.18 GB |
| | | Average time per batch | $3.52 \pm 0.17$ ms | $10.27 \pm 0.22$ ms | $7.00 \pm 0.12$ ms |
| ResNet-50 | ViT-L-14 | Peak GPU memory | 2.96 GB | 3.44 GB | 3.20 GB |
| | | Average time per batch | $5.85 \pm 0.11$ ms | $12.52 \pm 0.21$ ms | $9.43 \pm 0.19$ ms |
| ViT-B-16 (LAION-400M) | ViT-B-16 (LAION-2B) | Peak GPU memory | 1.46 GB | 2.29 GB | 1.88 GB |
| | | Average time per batch | $3.52 \pm 0.17$ ms | $9.18 \pm 0.12$ ms | $6.33 \pm 0.07$ ms |
| ViT-B-16 (LAION-2B) | ViT-L-14 (LAION-2B) | Peak GPU memory | 2.96 GB | 3.79 GB | 3.38 GB |
| | | Average time per batch | $5.85 \pm 0.11$ ms | $11.73 \pm 0.15$ ms | $8.89 \pm 0.11$ ms |

Table 2: Memory usage and average time per batch in inference with batch size 128.

| Source Models | Average Time per Token (ms) | | |
| --- | --- | --- | --- |
| | **Llama2-7B** | **Llama3.2-1B** | **Llama3.2-3B** |
| Target Models | **Llama2-13B** | **Llama3-8B** | |
| Pretrained | $26.0_{\pm 0.2}$ | $17.1_{\pm 1.1}$ | |
| EFT | $39.8_{\pm 0.1}\ (\times\ 0.65)$ | $24.4_{\pm 0.2}(\times 0.7)$ | $30.4_{\pm 0.0}(\times 0.56)$ |
| PRT | $27.8_{\pm 0.5}\ (\times\ 0.93)$ | $22.7_{\pm 1.8}(\times 0.75)$ | $23.1_{\pm 1.0}(\times 0.74)$ |

Table 3: Inference speed of Pretrained, EFT and PRT. The number following "×" in brackets indicates each method's token generation speed relative to the Pretrained model, whose speed is set to 1.0.

## D. Qualitative Analysis

To analyze the behavior of PRT in more detail, we examine the tokens generated by the model. In this analysis, we adopt Llama3-8B as the target model. In this analysis, we used Llama3-8B as the target model, while Llama-3.2-1B was used for training and reward-model initialization. Figures 8 and 9 present the output obtained under these settings.

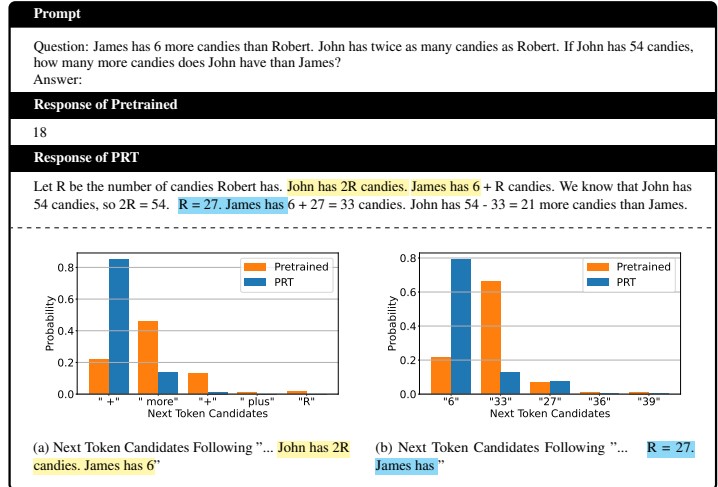

Figure 8: Example of model response.

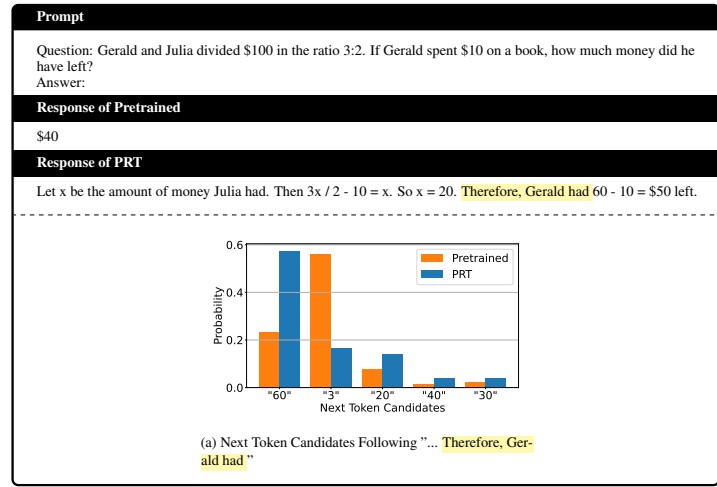

Figure 9: Example of model response.

# E. Ablation Study: Impact of Source Model Choice on PRT

In this section, we investigate the impact of source model differences on PRT by comparing GSM8K and IFEval scores when using Llama 3.2-1B and 3B, Qwen 2.5-0.5B and 1.5B, and Falcon 3-1B and 3B as source models. All models were trained under the same conditions using the training data and hyperparameters specified in Appendix B. The results for each case are shown in Figures 10 and Figures 11.

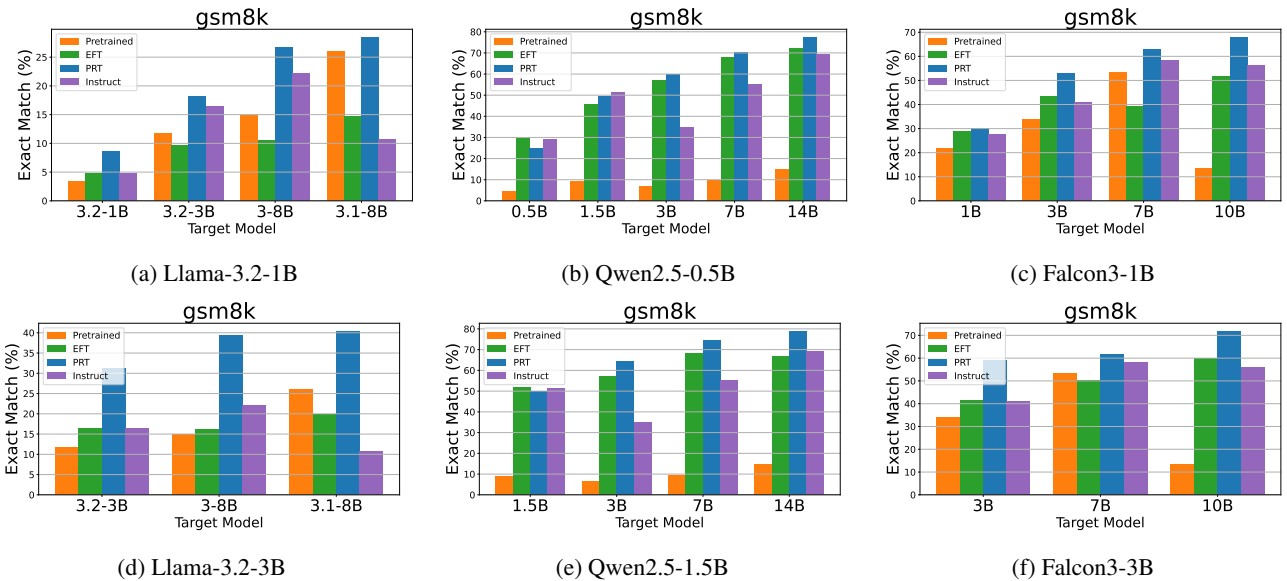

Figure 10: Evaluations of inference-time instruction-tuned models on the GSM8k benchmark. Each subcaption refers to the source pretrained model, and the labels in x-axis are target pretrained models. **Pretrained** means the zero-shot inference by each target model as a baseline, and **Instruct** means the instruct-tuned target model as an oracle result.

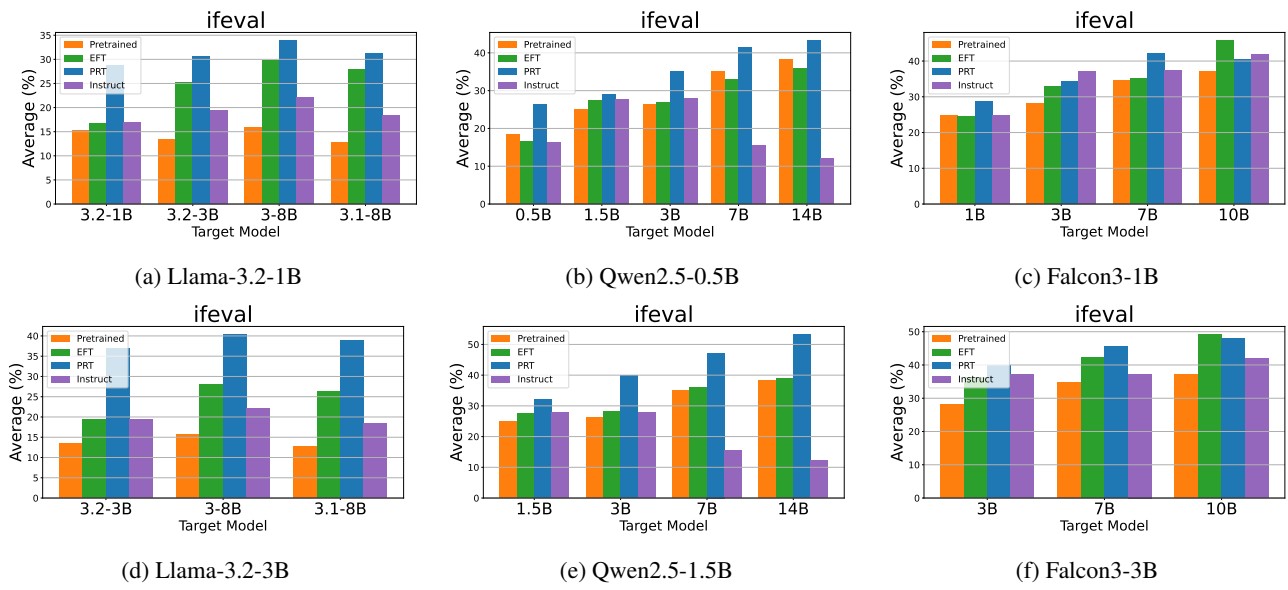

Figure 11: Evaluations of inference-time instruction-tuned models on the IFEval benchmark. Each subcaption refers to the source pretrained model, and the labels in x-axis are target pretrained models. **Pretrained** means the zero-shot inference by each target model as a baseline, and **Instruct** means the instruct-tuned target model as an oracle result.

## F. Ablation Study: Instruction-Tuning with Multiple Random Seeds

We conducted additional experiments using the Qwen-2.5 0.5 B model as the source model, where three independent training runs are carried out with different random seeds. As summarized in Table 4 the variance of evaluation scores across seeds is nearly identical between our method (PRT) and standard fine-tuning, indicating that PRT is as stable as standard fine-tuning for language-model training.

|     | 0.5B | 1.5B | 3B | 7B | 14B |
| --- | --- | --- | --- | --- | --- |
| EFT | $26.79 \pm 3.79\%$ | $45.94 \pm 4.40\%$ | $53.37 \pm 4.77\%$ | $66.14 \pm 2.43\%$ | $71.01 \pm 3.16\%$ |
| PRT | $26.69 \pm 1.45\%$ | $51.73 \pm 1.84\%$ | $62.37 \pm 2.14\%$ | $71.34 \pm 0.60\%$ | $77.23 \pm 0.20\%$ |

Table 4: Performance comparison of EFT and PRT with standard deviations across model sizes.

## G. Ablation Study: Robustness to Distribution Shifts in Inputs

Here we examine the robustness of PRT to input distribution shifts. For this purpose, we conducted experiments on CIFAR100 with Gaussian noise. We employ a reward model (ResNet50, untuned) trained on clean data by PRT, and then additionally tune it by PRT on the noisy data (ResNet50, tuned) for only a few iterations. In Table 5, we observed that (1) PRT degrades its performance on noisy data (as well as standard FT), but (2) additional PRT training can recover its performance.

|     | ResNet50 | ResNet101 | ViT-B-16 |
| --- | --- | --- | --- |
| PRT on clean data (ResNet50, untuned) | 71.70% | 72.36% | 79.5% |
| PRT on **noisy** data (ResNet50, untuned) | 21.37% | 34.77% | 53.47% |
| PRT on **noisy** data (ResNet50, **tuned**) | 71.60% | 72.18% | 78.06% |

Table 5: Performance of PRT under clean and noisy training conditions across different backbone models

## H. Ablation Study: PRT with LoRA

Here we conducted additional experiments of instruction-tuning by PRT with LoRA (Hu et al., 2022). Table 6 presents the results of instruction-tuned models with LoRA, evaluated on GSM-8k, showing that LoRA actually works with PRT as well as FT.

|     | 0.5B | 1.5B | 3B | 7B | 14B |
| --- | --- | --- | --- | --- | --- |
| EFT+LoRA (0.5B) | 29.72% | 53.15% | 65.28% | 56.41% | 53.37% |
| PRT+LoRA (0.5B) | 21.83% | 50.11% | 63.15% | 74.91% | 73.62% |

Table 6: Comparison of EFT+LoRA and PRT+LoRA across model sizes

## I. Empirical Evaluation for $\varepsilon$ and $C$

We conducted additional experiments to measure KL-divergence $\varepsilon$ between pretrained models and the constant $C$ appeared in Proposition 3.2. In Table 7, we reported averaged results over inputs from the dataset. These results indicate that (1) KL divergences between similar models are small as expected, (2) KL divergences are affected by differences in pretraining datasets, rather than model architectures, and (3) the constant C is finitely bounded in this setting.

## J. Evaluation on Various Vision Tasks

Figure 12 shows an extensive evaluation of inference-time tuning on various vision datasets.

| $\pi_{pt}$ | $\widetilde{\pi}_{pt}$ | $\varepsilon$ | $C$ |
|---|---|---|---|
| RN50 (OpenAI) | RN50 (OpenAI) | 0.0 | 19.41 |
| RN50 (OpenAI) | RN101 (OpenAI) | 0.0016 | – |
| | ViT-B (OpenAI) | 0.0030 | – |
| | ViT-B (LAION-400M) | 0.0107 | – |
| | ViT-L (OpenAI) | 0.0044 | – |
| | ViT-L (LAION-400M) | 0.0158 | – |

Table 7: Empirical Evaluation of $\varepsilon$ and $C$ in Proposition 3.2.

## K. Evaluation on Code Generation Tasks

To evaluate the code generation capabilities of PRT, we conducted a comparison using HumanEval (Chen et al., 2021).

**HumanEval** : A dataset for evaluating the ability of models to generate correct code based on natural language descriptions.

**Result** As shown in Figure 13, the effectiveness of PRT was confirmed for Llama2, Llama3, and Qwen2.5. On the other hand, for Falcon3 with target model sizes of 7B and 10B, the score actually declined compared to the Pretrained model. We believe this is because, during training, the same 1B model was used as the target, which exhibited little improvement over the Pretrained model in code generation. As a result, the PRT model did not acquire code generation capabilities during training; consequently, when applied to larger model sizes, it acts as noise rather than providing a benefit.

## L. Evaluation on Various NLP Tasks

To investigate the effectiveness of PRT, we conducted comparisons using a diverse set of evaluation datasets. In addition to GSM8k, IFEval, and HumanEval, we used the following datasets for evaluation.

- **Academic / General Knowledge MCQ**

    - **ARC** (Clark et al., 2018): The AI2 Reasoning Challenge (ARC) dataset is designed to test a model's ability to answer grade-school level science questions.
    - **MMLU** (Hendrycks et al., 2020): The Massive Multitask Language Understanding (MMLU) benchmark tests a model's ability to perform a wide range of language understanding tasks.
    - **MMLU Pro** (Wang et al., 2024c): An extension of the MMLU benchmark with more challenging tasks.

- **Code Generation / Programming**

    - **MBPP** (Austin et al., 2021): The Mostly Basic Programming Problems (MBPP) dataset is designed to test a model's ability to solve basic programming problems.

- **Commonsense Reasoning**

    - **Hellaswag** (Zellers et al., 2019): A dataset for evaluating commonsense reasoning and natural language inference.
    - **GPQA** (Rein et al., 2023): A dataset for evaluating general-purpose question answering capabilities of models.

- **Truthfulness**

    - **TruthfulQA** (Lin et al., 2021): A dataset for evaluating the truthfulness of answers generated by models.

In Figure 14, we compared PRT with the baseline using the Llama2 series on these benchmarks. Each method employs the same models used in the NLP experiments in Section 4, which were trained with Tulu v2.

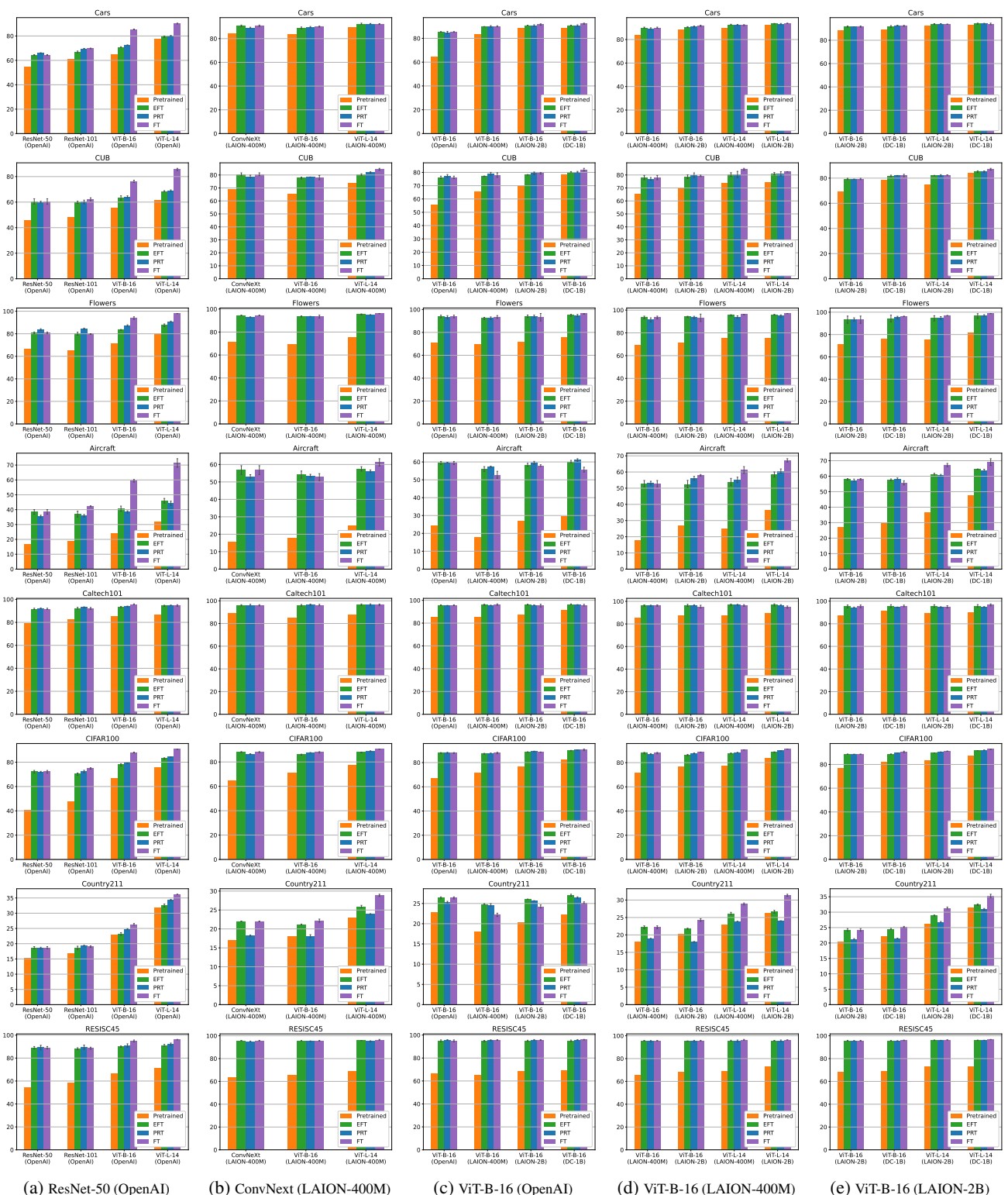

Figure 12: Evaluations of inference-time tuned models for vision tasks. Each subcaption refers to the source pretrained model, and the labels in x-axis are target pretrained models. **Pretrained** means the zero-shot classification by each target model as a baseline, and **FT** means the fine-tuned target model as an oracle result.

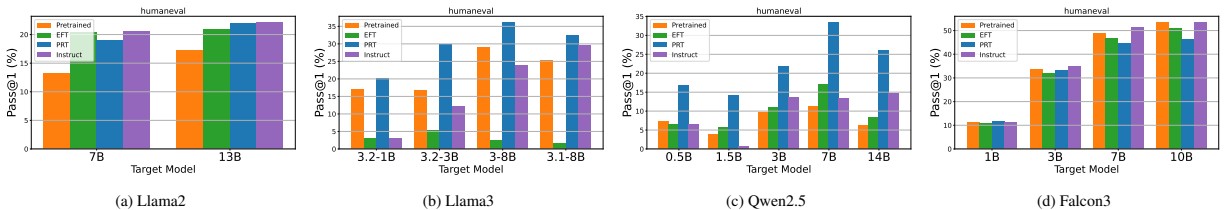

Figure 13: Comparison of PRT and EFT on HumanEval.

(a) Llama2  (b) Llama3  (c) Qwen2.5  (d) Falcon3

(a) ARC Easy  (b) ARC Challenge  (c) GPQA Diamond

(d) GSM8k  (e) Hellaswag  (f) HumanEval

(g) IFEval  (h) MBPP  (i) MMLU

(j) MMLU PRO  (k) TruthfulQA

Figure 14: Evaluations of inference-time instruction-tuned models on Llama 2 Series. Each subcaption refers to the task name, and the labels in x-axis are target pretrained models. **Pretrained** means the zero-shot inference by each target model as a baseline, and **Instruct** means the instruct-tuned target model as an oracle result.

