# OpenReview forum: "Portable Reward Tuning: Towards Reusable Fine-Tuning across Different Pretrained Models"
_ICML.cc/2025/Conference — ICML 2025 poster_

### Official Review · Reviewer_RJqU · 2025-03-11

**Overall Recommendation:** 3

**Summary:**

The paper proposes Portable Reward Tuning (PRT), a fine-tuning pipeline to enable efficient and reusable fine-tuning across different foundation models (FMs). It is especially useful when the old FMs are replaced by new FMs with different pre-training dataset and even different model architectures. Previous fine-tuning methods, i.e. inference-time tuning methods or emulated fine-tuning (EFT) in this paper, require running multiple models during inference, increasing overhead. The proposed PRT claims to avoid this by training an explicit reward model instead of modifying the pretrained model for each fine-tuning task. Experiments on vision and language models show that PRT achieves comparable accuracy to EFT with lower inference costs (evaluated by memory usage and time usage).

**Claims And Evidence:**

As my understanding, with the proposed PRT method, this paper makes three claims and provides proof to support each claim.
Claim 1: PRT maintains accuracy comparable to EFT. Claim 2: PRT reduces inference overhead compared to EFT. Claim 3: During inference, the reward model can be used with any foundation model (with the same set of vocabularies or labels).
For Claim 1, Figures 2–4 and Appendix E,F show PRT matches EFT on vision and language tasks. For Claim 2, the authors provide Tables 2-3 in Appendix. For Claim 3, the above experiment results can prove it. For example, for CLIP, we see generalization across ResNet and ViT, for LLAMA we see results from 1B model to 8B model, and for Qwen (from 0.5 B to 72 B), etc.

**Essential References Not Discussed:**

To my knowledge, the current related work is comprehensive.

**Experimental Designs Or Analyses:**

1. Quantitative experiments on Vision experiments and Language experiments
a) The settings of baselines. For each experiment, the comparison is limited within FT, EFT and PRT, and there are other lightweight efficient fine-tuning methods available as baseline methods, such as LoRA and adapters. It weakens the point that PRT is a superior alternative to fine-tuning.
b) FT method performs better than EFT and PRT in terms of accuracy and speed in most of the cases (Fig. 2 and 10. Table 1 and 2), yet FT is not a baseline. The impact is the same with the last issue.
c) About the training and inference speed, the proof is provided in the Appendix. I feel it is also an important proof because it is one of the keys to proving the efficiency of the proposed method. Is it possible to have them or part of them in Section 4?

2. Qualitative experiments
a) It lacks analysis of Figure 11 - 12 in HumanEval Results (Appendix F). For example, PRT underperforms on Falcon-3 models when the target model is 7B and 10 B. Things are different with other target models. What does this phenomenon mean? Does it imply that PRT’s reliance on source model quality, which is not discussed in the main paper?


3. Other issues not discussed in the paper
a) The reward model scalability. What factors can impact the scale of the reward model and how costs would change with its scale?
b) What are the limits of the "same label space" assumption? And what if the label is partially different?

**Methods And Evaluation Criteria:**

The proposed methods: PRT trains a reward model using cross-entropy loss, reformulating fine-tuning as reward maximization with KL regularization. The reward is combined with new models at inference via closed-form policy (Eq. 10).

The experiments are designed on broad coverage of FMs and benchmarking datasets. The paper provides evaluation on proposed method, baseline methods (EFT) and FT for both vision tasks and language tasks. However, I believe the evaluation metrics are not sufficient. For example, for vision tasks, only one quantitative metric, accuracy (Figs. 2–4) is used. The accuracy focuses on means with no standard deviations or confidence intervals. Is it possible to have more metrics to provide statistical significance of differences between PRT and baselines?

**Other Comments Or Suggestions:**

1. The ‘flower’ dataset used for experiments in Figure 2 is not mentioned in section 4 but only appears in the Appendix.
2. Section 4.4 ‘Tables 2 and 3 show that PRT successfully reduces both inference speed and memory usage’ should be ‘...increase inference speed’ (increase speed or reduce time).
3. For Table 1, and Table 2 in the Appendix, ‘Speed per batch’ should be ‘Time usage per batch’.

**Other Strengths And Weaknesses:**

S1. The paper provides an interesting idea to bridge an old FM and a new FM with a tuned reward, without training from the beginning every time for a new FM.
W1. However I believe the motivation of using PRT is not persuasive enough because 1) the direct FM performs well. 2) Other efficient fine-tuning methods are not compared to prove the high efficiency of PRT.

**Questions For Authors:**

Q1. We have seen PRT is capable of handling different datasets and even model architectures. Is it possible to test PRT method on different downstream tasks, and how to do that? For example, for vision tasks, there are different downstream tasks including object detection, image segmentation, etc.

Q2.  Is it possible to have a comparison of PRT with other latest effective fine-tuning methods?

Q3. Could the trained reward model of PRT be ‘outdated’ and require re-training too?

**Relation To Broader Scientific Literature:**

This work, if solid, may contribute to Foundation Models of both computer-vison and natural language processing.

**Theoretical Claims:**

Following are theoretical claims I have checked.

Proposition 3.1: Establishes a one-to-one mapping between fine-tuned models and rewards. Comment: Valid, assuming rewards are scaled properly.

Proposition 3.2: Bounds the KL divergence between inference models when source and target pre-trained models are ε-close.
Comment: 1) When replacing a Resnet foundation model with ViT (just like the experiment does. See Figure 2 (a)), the ε should be larger than replacing a Resnet with a Restnet. I am not sure how drastic the ε could be in practice. 2) The authors assume that the maximum and mean value ratio of the exponential reward is bounded by some constant C. I am not sure how large the C could be and if this assumption is realistic for practice or not.

Proposition 3.3: I am not familiar with the PAC Bayes framework so I lack understanding of this.

---

> ### Author Rebuttal · Authors · 2025-04-01
>
> Thank you for your detailed feedback. Here we focus on answering major concerns/questions due to space limitation. However, we sincerely appreciate other feedback and will reflect them to our revisions, as well as discussions below.
>
> ### Methods And Evaluation Criteria
> > However, I believe the evaluation metrics are not sufficient.
>
> Due to the space limitation, please refer to our answer to Reviewer BmQN on standard deviations.
>
> ### Theoretical Claims
> > When replacing a Resnet foundation model with ViT, the ε should be larger than replacing a Resnet with a Restnet. I am not sure how drastic the ε could be in practice.
>
> > I am not sure how large the C could be and if this assumption is realistic for practice or not.
>
> The practical validity of our assumptions is indeed an important perspective. Thus we conducted additional experiments to measure $\epsilon$, KL-divergence between pretrained models, and the constant $C$ on CIFAR100:
>
> |$\pi_{pt}$|$\widetilde{\pi}_{pt}$|ε|C|
> |-|-|-|-|
> |RN50 (OpenAI)|RN50 (OpenAI)|0.0|19.41|
> |RN50 (OpenAI)|RN101 (OpenAI)|0.0016|-|
> ||ViT-B (OpenAI)|0.0030|-|
> ||ViT-B (LAION-400M)|0.0107|-|
> ||ViT-L (OpenAI)|0.0044|-|
> ||ViT-L (LAION-400M)|0.0158|-|
>
> Here we reported averaged results over inputs from the dataset. These results indicate that (1) KL divergences between similar models are surprisingly small, (2) KL divergences largely depend on pretraining datasets, rather than model structures, (3) the constant C is seemingly finitely bounded.
>
> ### Experimental Designs Or Analyses
> > [...] there are other lightweight efficient FT methods available as baseline methods, such as LoRA and adapters. It weakens the point that PRT is a superior alternative to FT.
>
> > FT method performs better than EFT and PRT in terms of accuracy and speed in most of the cases, yet FT is not a baseline. The impact is the same with the last issue.
>
> First of all, we would like to emphasize that our paper **does not** claim PRT is `a superior alternative to FT` unconditionally. If we have enough resources that we can ignore costs for training models or maintaining training data, we should employ repeated FT rather than one-time PRT. Rather, our paper claims that PRT is a better alternative in cases that we don't want to retrain new pretrained models for various reasons. In such cases, FT models are unavailable and thus should be viewed as unknown oracles for PRT/EFT. Hence **the fact that FT is better than PRT/EFT in accuracy/efficiency does not diminish their motivation.**
>
> Also, we consider that the lack of comparison with efficient FT methods like LoRA **does not weaken our claims** because: (1) Both PRT and EFT are not efficient FT methods. Rather, they enable us to reuse a once-tuned model for (inference-time) tuning other pretrained models with different sizes or updated knowledge. (2) Since efficient FT methods like LoRA are designed to apply to FT, they should also apply to PRT. The following additional experiments (instruct-tuned with LoRA, eval. on GSM8K) verify that LoRA actually works with PRT as well as FT.
>
> ||0.5B|1.5B|3B|7B|14B|
> |-|-|-|-|-|-|
> |EFT+LoRA (0.5B)|29.72%|53.15%|65.28%|56.41%|53.37%|
> |PRT+LoRA (0.5B)|21.83%|50.11%|63.15%|74.91%|73.62%|
>
> > It lacks analysis of Figure 11-12 in HumanEval Results (...). For example, PRT underperforms on Falcon-3 models when the target model is 7B and 10B. Things are different with other target models. [...]
>
> Actually, PRT performs well on the GSM8K benchmark even with Falcon3 7B/10B models, which implies PRT indeed transfer the instruction-tuned ability to these models. The problem on HumanEval is mainly due to the difficulty of controlling downstream accuracy in instruction-tuning in some models, since the instruction dataset contains many data irrelevant to downstream tasks.
>
> > Other issues not discussed in the paper a) The reward model scalability. [...] b) What are the limits of the "same label space" assumption? And what if the label is partially different?`
>
> a): Figure 13 shows the reward model scalability. Overall, a larger reward model leads to better accuracy with new pretrained models. Also, we additionally evaluated how inference time will be changed by scailing model sizes (1B → 3B). The results show the increase in average time per token is only 2% with PRT, while 25% with EFT.
>
> ||Llama3.2-1B w/ Llama3-8B|Llama3.2-3B w/ Llama3-8B|
> |-|-|-|
> |EFT|24.4 ± 0.2 ms|30.4 ± 0.0 ms (× 1.25)|
> |PRT|22.7 ± 1.8 ms|23.1 ± 1.0 ms (× 1.02)|
>
> b): See our answer to Reviewer BmQN.
>
> ### Questions:
> Q1: The instruction-tuned models are actually evaluated on downstream tasks like math and coding. However, as you pointed out, tasks that require feature extraction, rather than label distributions, are out of scope in this paper. This may be an interesting direction for future work.
>
> Q2: See above discussion.
>
> Q3: We confirmed that the reward model can be retrained and then reused with other pretrained models when input distribution changed. See our answer to Reviewer BmQN.

---

> > ### Comment · Reviewer_RJqU · 2025-04-07
> >
> > Thanks for the detailed responses. The rebuttal is persuasive and add more details based on the original submission. I updated my rating accordingly.

---

> > > ### Author Response · Authors · 2025-04-08
> > >
> > > Thank you for taking the time and updating your evaluation. We again appreciate your detailed feedback, which led to the improved manuscript and a better understanding of our method.

---

### Official Review · Reviewer_BmQN · 2025-03-13

**Overall Recommendation:** 3

**Summary:**

This paper proposes Portable Reward Tuning (PRT), a new fine-tuning paradigm that decouples the “reward” from the foundation model itself, thereby making it “portable” to other foundation models of the same architecture family (with shared label or token vocabulary). Overall, the method aims to reduce repeated training costs and extra inference costs when one’s underlying foundation model is replaced or upgraded.

**Claims And Evidence:**

Overall, the paper offers sound conceptual reasoning plus decent experiments on a variety of image and NLP tasks. While the final accuracy typically remains below a fully re-fine-tuned new model, PRT competes well with the existing “emulated fine-tuning” baseline and indeed uses fewer inference resources.

**Essential References Not Discussed:**

No major references are obviously missing.

**Experimental Designs Or Analyses:**

The paper’s results cover classification (eight standard fine-grained or broad datasets), instruction following tasks (GSM8k, IFEval, MMLU, etc.), and code generation (HumanEval). This breadth is good.

The authors only do a certain limited ablation around, e.g., the effect of different source vs. target architectures or training stable reward networks. More discussions on how stable the training is across random seeds or large scale might be helpful.

**Methods And Evaluation Criteria:**

The authors design PRT by explicitly parameterizing a reward function r_\theta(x,y). They then train it via the same cross-entropy style objective normally used for fine-tuning, ensuring that the closed-form “max reward + KL constraint” policy solution matches the fine-tuned distribution. They measure classification accuracy in vision tasks, code generation pass rates, or language understanding metrics. The baselines are: (1) the original, non-fine-tuned “pretrained” model, (2) a normal “fine-tuned” model for that new architecture (oracle), (3) the existing “emulated fine-tuning” approach (EFT), and (4) PRT.

The evaluation is thorough for a wide set of classification and NLP tasks. However, exploring truly large language models in real production settings or more open-ended tasks would bolster real-world relevance.

**Other Comments Or Suggestions:**

Could we do something approximate if the new model’s vocabulary is only partly changed? Another possible future direction is combining PRT with partial re-training to reduce any performance gap.

**Other Strengths And Weaknesses:**

Weaknesses: 	Some details about how robust the reward model is to massive distribution shifts remain underexplored. And the final performance is typically still below a real full re-fine-tune. If one can afford re-training, PRT is only partial.

Overall, it’s a promising approach.

**Questions For Authors:**

How big a mismatch can we handle between old and new model vocabularies or label sets? Could partial alignment or token mapping methods mitigate that? Do you observe any stability concerns or require special hyperparameters to get consistent results in PRT training?

**Relation To Broader Scientific Literature:**

Fine-tuning with KL constraints is widely used in RL from human feedback. The authors cite prior works that interpret instruction tuning or RLHF in such terms. The authors position PRT as a solution for “cross-model generalization of fine-tuned solutions.” They might connect more with methods in model distillation or universal prompt offsets, but that is less critical.

**Theoretical Claims:**

The authors rely on the standard derivation of KL-regularized maximum entropy RL. They show that “fine-tuning is the closed-form solution to a certain reward objective.” They provide a fairly standard PAC-Bayesian argument in the appendix to justify generalization.

---

> ### Author Rebuttal · Authors · 2025-04-01
>
> Thank you for your valuable feedback. We are really encouraged by the positive feedback to our research direction. Here we would like to address your concerns or questions.
>
> > The authors only do a certain limited ablation around, e.g., the effect of different source vs. target architectures or training stable reward networks. More discussions on how stable the training is across random seeds or large scale might be helpful.
>
> Thank you for the suggestion. Indeed, in vision experiments (e.g. Fig 2), we have already plotted one standard deviation (by black bars) over three random seeds in training, which indicates very small variance in PRT training/inference. Moreover, although training language models with multiple seeds is computationally heavy and thus not standard in previous literature, we additionally conducted PRT/Instruct tuning of Qwen2.5-0.5B model with three random seeds, and evaluated them with other pretrained models. These results show the stability of PRT is at the same level as standard FT, also in language experiments. We would like to add such discussions on training stability with respect to random seeds in our revision.
>
> ||0.5B|1.5B|3B|7B|14B|
> |---|---|---|---|---|---|
> |EFT|$26.79\pm3.79\%$|$45.94\pm4.40\%$|$53.37\pm4.77\%$|$66.14\pm2.43\%$|$71.01\pm3.16\%$|
> |PRT|$26.69\pm1.45\%$|$51.73\pm1.84\%$|$62.37\pm2.14\%$|$71.34\pm0.60\%$|$77.23\pm0.20\%$|
>
> > The authors position PRT as a solution for “cross-model generalization of fine-tuned solutions.” They might connect more with methods in model distillation or universal prompt offsets, but that is less critical.
>
> Thank you for suggestions. We agree that automatic prompt tuning can also be considered as inference-time tuning (but only for language models), and thus we would like to add a discussion about them. Also we would like to survey related literatures from the field of model distillation.
>
> > Weaknesses: Some details about how robust the reward model is to massive distribution shifts remain underexplored.
>
> > [...] Another possible future direction is combining PRT with partial re-training to reduce any performance gap.
>
> Here we will proceed by assuming `massive distribution shifts` refers to input distributions. To examine this, we conducted experiments with noisy data, i.e., CIFAR100 with gaussian noise ($\sigma^2=0.01$). Here we employ a reward model (ResNet50, untuned) trained on clean data by PRT, and then additionally tune it by PRT on the noisy data (ResNet50, tuned) for only $1/10$ iterations. The results show that (1) PRT degrades its performance on noisy data (as well as standard FT) but (2) additional PRT training can recover its performance.
>
> |           |ResNet50|ResNet101 |ViT-B-16|
> |---|---|---|---|
> |PRT on clean data (ResNet50, untuned)       |71.70%|72.36%|79.5%|
> |PRT on **noisy** data (ResNet50, untuned)       |21.37%|34.77%|53.47%|
> |PRT on **noisy** data (ResNet50, **tuned**)|71.60%|72.18%|78.06%|
>
> > Could we do something approximate if the new model’s vocabulary is only partly changed?
> > How big a mismatch can we handle between old and new model vocabularies or label sets? Could partial alignment or token mapping methods mitigate that?
>
> We sincerely agree that such a research direction is important for inference-time tuning, i.e., for both EFT and PRT. We decided not to treat this topic in this PRT paper as it should be focused on questions specific to PRT. Also handling different vocabularies is actually a highly non-trivial problem in language models [1, 2, for e.g.], and thus it should deserve another paper, which may require to match different vocabularies using, for e.g., edit distance, optimal transport, etc.
> - [1] Xu et al. "Bridging the Gap between Different Vocabularies for LLM Ensemble"
> - [2] Wan et al. "Knowledge Fusion of Large Language Models"
>
> > Do you observe any stability concerns or require special hyperparameters to get consistent results in PRT training?
>
> As described in Appendix B, we basically used the same hyperparameters for both PRT and (E)FT in training and inference, for fair comparison. Some experiments (Fig.6, 7) includes EM regularization, but we found that the regularization coefficient does not affect the stability of final results.

---

> > ### Comment · Reviewer_BmQN · 2025-04-05
> >
> > After reviewing the authors’ rebuttal, I appreciate the additional experiments. Given my limited expertise in this area, I maintain my original score, and the AC may weigh my evaluation accordingly.

---

> > > ### Author Response · Authors · 2025-04-05
> > >
> > > We sincerely thank you for taking the time to read our rebuttal and for acknowledging the additional experiments. As our paper appears to be on the borderline, we would greatly appreciate it if you could consider reassessing your evaluation in light of our clarifications to your concerns, or kindly let us know if there are any remaining concerns we could address in future work.
> > >
> > > Once again, we really appreciate your time and thoughtful review!

---

### Official Review · Reviewer_8QF5 · 2025-03-16

**Overall Recommendation:** 4

**Summary:**

This paper introduces a novel approach to reward tuning that can be transferred across model architectures. Instead of modifying a model parameters directly, the proposed approach trains an explicit reward model using the same objective as fine-tuning. At inference time, the reward model can be applied to any compatible foundation model without additional retraining. Experiments on vision and language tasks demonstrate that the proposed algorithm achieves comparable accuracy to traditional fine-tuning.

**Claims And Evidence:**

The authors conduct convincing experiments and comparisons to support the strengths of their proposed algorithm.

**Essential References Not Discussed:**

The authors have extensively discussed the important references.

**Experimental Designs Or Analyses:**

The experiment designs are sound and valid, relevant to the claims made in the paper.

**Methods And Evaluation Criteria:**

The experimental settings selected in the paper clearly support the claims made by the authors in the paper.

**Other Comments Or Suggestions:**

I don't find any typos.

**Other Strengths And Weaknesses:**

The strength of the paper lies in its motivation to make fine-tuning more efficient by introducing Portable Reward Tuning (PRT), which allows reward models to be reused across different pretrained models. The authors carefully discuss the key principles behind their approach, framing fine-tuning as reward maximization with KL regularization, which reduces inference costs while maintaining performance. Additionally, the paper provides theoretical insights into how reward models adapt to different pretrained models. Overall, this paper presents a promising direction for making model adaptation more efficient and reusable.

Please find my questions below.

**Questions For Authors:**

There are a few questions that I would like the authors to clarify during the rebuttal.

- There is a mismatch between algorithm 1 and the text in lines 171-206. Reward models are simply fine-tuned on a given task with cross entropy loss. However, algorithm 1 seems to be training a reward model that trains on the difference between pre-training model's predictions and the true distribution from the task. In the first scenario, the reward function that will be used in algorithm 2 will be a log ratio of the fine-tuned model's prediction and original pre-trained model's prediction. On the other hand, in second scenario, the reward model can be used directly. Please clarify which version is used in the experiments. Furthermore, please clarify how many models need to be maintained to perform PRT at inference?

- In figure 3, why is Instruct's performance lower than others? Isn't Instruct a model that has been fine-tuned on the task itself? Please clarify the differences between FT and Instruct in figures 2 and 3.

- In proposition 3.1, when defining r(y|x), shouldn't $\pi_{\text{ft}}$ be utilized for sampling y, i.e. $E_{y \sim \pi_{\text{ft}}} r(x, y) = 1$?

- How is the regularization term applied? Is it simply applying an entropy regularization on the log likelihood ratio between the pre-trained model and the model that is being fine-tuned? What are the differences of this regularization term from the KL divergence loss between the pre-trained model and fine-tuned model's predictions?

**Relation To Broader Scientific Literature:**

The paper studies an important question on adaptable and reusable reward models that can utilized across pre-trained models. The proposed algorithm is simple, with a theoretical basis.

**Theoretical Claims:**

I quickly glanced the proofs for the theoretical claims to  support the design of their algorithm and they look reasonable.

---

> ### Author Rebuttal · Authors · 2025-04-01
>
> Thank you for your valuable feedback, and finding our research direction promising. Here we would like to address the questions.
>
> > There is a mismatch between algorithm 1 and the text in lines 171-206. Reward models are simply fine-tuned on a given task with cross entropy loss. However, algorithm 1 seems to be training a reward model that trains on the difference between pre-training model's predictions and the true distribution from the task.
>
> Sorry for the confusion. We reviewed this part and confirmed there is actually no mismatch. Here let us explain that. From the text, we wrote `The reward model $r_\theta(x,y)$ is trained by simply optimizing the same loss function $L(p, y^*)$ as in standard fine-tuning` and the loss function is expanded as $L(p, y) = ... = -\log \pi_\theta (y | x)$. This description corresponds to line 8 in Algorithm 1. Also importantly, we note that $\pi_\theta (y|x)$ is defined in eq. (6), as the product of the pretrained model $\pi_\mathrm{pt}(y|x)$ and the exponential of the reward model $r_\theta(x,y)$. This corresponds to lines 6-7 in Algorithm 1, where we compute the product in the logarithmic space and then take softmax to obtain $\pi_\theta (y|x)$. Nevertheless, as these correspondences are not explicitly described in the text, we would like to add such annotations in our revised paper. Thank you for pointing out them.
>
> > In the first scenario, the reward function that will be used in algorithm 2 will be a log ratio of the fine-tuned model's prediction and original pre-trained model's prediction. On the other hand, in second scenario, the reward model can be used directly. Please clarify which version is used in the experiments. Furthermore, please clarify how many models need to be maintained to perform PRT at inference?
>
> (cont.) Here we note that our reward model is expected to play the same role as the log ratio of the two models. In other words, Algorithm 1 automatically learns a single model $r_\theta(x,y)$ that implements this role, and that is the point of PRT. So the answer for the last question is: two models (new pretrained model & reward model) are used in PRT inference, while three models (new pretrained model & old pretrained model & old FT model) are used in the previous work.
>
> > In figure 3, why is Instruct's performance lower than others? Isn't Instruct a model that has been fine-tuned on the task itself? Please clarify the differences between FT and Instruct in figures 2 and 3.
>
> First of all, let us clarify the differences between FT (in vision experiments) and Instruct (in language experiments). FT means a model fine-tuned and evaluated on the same expert dataset (e.g., Cars, CUB, etc). On the other hand, Instruct means a model fine-tuned on a general dataset (i.e., the instruction-following data) and evaluated on **other** expert datasets like math and coding. So the answer for the second question is yes. Thus, the performance of Instruct on each expert benchmark may degrade on some dataset due to implicit bias occurred in insturction tuning.
>
> > In proposition 3.1, when defining r(y|x), shouldn't be utilized for sampling y, i.e. $E_{y\sim \pi_{\mathrm{ft}}} r(x,y) = 1$?
>
> Sorry for the confusion. Thanks to this comment, we have noticed a (non-crucial) typo. The probability for sampling y, $\pi_{\mathrm{pt}}$, is actually correct. However, we should write the condition for $r(x,y)$ should be $E_{y\sim \pi_{\mathrm{pt}}} \exp r(x,y) = 1$.
>
> Let us also explain about this condition briefly. By the mapping in proposition 3.1, $\pi_\mathrm{ft}$ is mapped to $r(x,y) := \log(\pi_\mathrm{ft}(y|x) / \pi_\mathrm{pt}(y|x))$. This satisfies $$E_{y\sim \pi_{\mathrm{pt}}} \exp r(x,y) = \sum_y \pi_{\mathrm{pt}}(y|x) \exp r(x,y) = \sum_y \pi_{\mathrm{pt}}(y|x) (\pi_{\mathrm{ft}}(y|x) / \pi_{\mathrm{pt}}(y|x)) = \sum_y \pi_{\mathrm{ft}}(y|x) = 1,$$
> i.e., $E_{y\sim \pi_{\mathrm{pt}}} \exp r(x,y) = 1$ rather than $E_{y\sim \pi_{\mathrm{ft}}} \exp r(x,y) = 1$.
>
> > How is the regularization term applied? Is it simply applying an entropy regularization on the log likelihood ratio between the pre-trained model and the model that is being fine-tuned? What are the differences of this regularization term from the KL divergence loss between the pre-trained model and fine-tuned model's predictions?
>
> The application of the regularization term is as described in eq. (12). The difference between the two KL divergence (eq. (5) and eq. (11)) is an important point. On one hand, KL divergence in eq. (5) measures discrepancy of two distributions over the label/vocabulary space. On the other hand, KL divergence in eq. (11) measures discrepancy of two (meta-)distributions over the set of probability models. As discussed in lines 267-274 and 220-229, since the latter KL divergence is computationally intractable, we proposed the entropy maximization regularization (eq. (5)).

---

> > ### Comment · Reviewer_8QF5 · 2025-04-08
> >
> > Hi,
> > Thank you for the detailed responses. I have increased my scores accordingly.

---

> > > ### Author Response · Authors · 2025-04-09
> > >
> > > Thank you very much for taking the time and for your positive evaluation. We also appreciate your detailed feedback in the initial review, which should make our manuscript more precise and easier to follow.

---

### Decision · Program_Chairs · 2025-05-01

**Decision:**

Accept (poster)

**Comment:**

This paper introduces Portable Reward Tuning (PRT), a new paradigm that reformulates fine-tuning as reward maximization with KL regularization. Rather than modifying the foundation model (FM) directly, the proposed method trains a reward model that can be reused across models with shared label or vocabulary spaces. This enables inference-time adaptation of newer FMs using previously learned reward models, reducing the need to retrain or run older models alongside new ones. This paper is a well-motivated, methodologically sound, and practically valuable contribution. I recommend acceptance.